# Symmetric Replay Training: Enhancing Sample Efficiency in Deep Reinforcement Learning for Combinatorial Optimization

## Abstract

Combinatorial optimization (CO) is frequently encountered in various industrial fields, such as drug discovery or hardware design. Despite its widespread relevance, solving CO problems is highly challenging due to the vast combinatorial solution space. Notably, CO problems in practice frequently necessitate computationally intensive evaluations of the objective function, further amplifying the difficulty. For efficient exploration with the limited availability of function evaluations, this paper introduces a new generic method to enhance sample efficiency. We propose symmetric replay training that leverages the high-reward samples and their under-explored regions in the symmetric space. In replay training, the policy is trained to imitate the symmetric trajectories of these high-rewarded samples. The proposed method is beneficial for the exploration of highly rewarded regions without the necessity for additional online interactions – *free*. The experimental results show that our method consistently improves the sample efficiency of various DRL methods on real-world tasks, including molecular optimization and hardware design. Our source code is available at https://anonymous.4open.science/r/sym_replay.

## 1 Introduction

Combinatorial optimization (CO) problems arise across diverse industrial domains, but they are notoriously challenging to solve. In CO (e.g., traveling salesman problems; TSP), the massive discrete solution space often leads to NP-hardness. Furthermore, CO problems in practical scenarios often involve computationally expensive objective functions to evaluate (e.g., a black-box function), introducing significant restrictions on the problem-solving process. Even though solving CO problems with expensive objective functions has broad impacts on the industry since they are frequently found in various fields like drug discovery or hardware design, it has numerous challenges.

Recently, deep reinforcement learning (DRL) has drawn significant attention to solve CO problems. The main promise of DRL algorithms is that they do not necessitate expert-designed labeled data or problem-specific knowledge to design solvers. They have shown impressive performance on various tasks (Olivecrona et al., 2017; Kool et al., 2018; Ahn et al., 2020; Park et al., 2023; Bengio et al., 2021; Kim et al., 2023). In general, utilizing extensive samples enhances training stability without introducing significant computational challenges. However, reward computation can be expensive in practice, so it becomes a computational bottleneck. For instance, reward computation usually contains simulations in black-box CO or another optimization problem in bi-level CO. In this regard, reducing the number of reward evaluations is more beneficial than stabilizing training with large batch size. This discrepancy raises a straightforward research question: *How can we enhance the sample efficiency of DRL methods for a broad spectrum of CO problems and methods?*

This paper proposes a new generic DRL method, *symmetric replay training (SRT)*, that leverages the solution-symmetric nature of combinatorial space. In DRL for CO, multiple action sequences (i.e., action trajectories) can be mapped to a single combinatorial solution, and the reward function is defined on the combinatorial solution space, not the action trajectory space. To efficiently explore the vast action space, we decompose the training process into two iterative steps: reward-maximizing training and symmetric replay training.

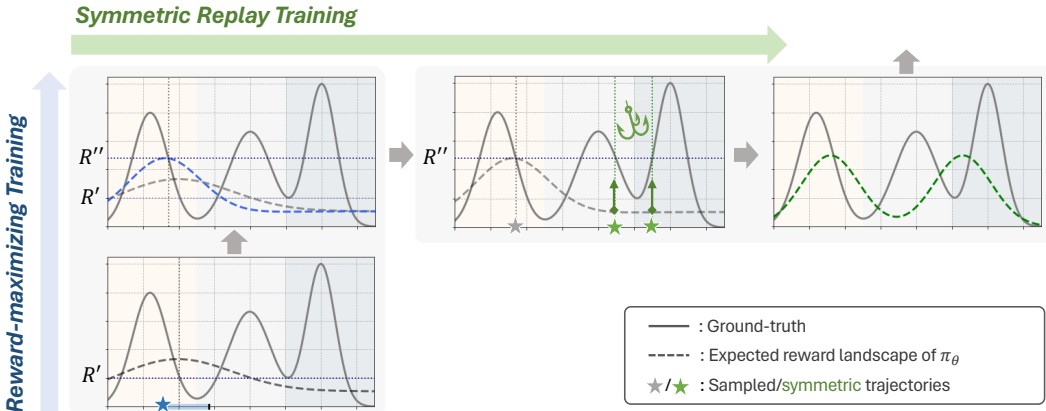

Figure 1: Illustration for two-step training strategies: reward-maximizing training and symmetric replay training.

Reward-maximizing training is conducted with a conventional DRL algorithm, which involves the exploration of the high-rewarded action trajectories over the entire action space. Subsequently, symmetric replay training bootstraps the policy in the symmetric space to enhance sample efficiency. In this process, the high-rewarded action trajectories obtained in the previous step are symmetrically transformed, and the policy is trained to imitate these symmetric action trajectories. As illustrated in Figure 2, the symmetric replay training promotes the exploration of under-explored regions containing trajectories equivariant to the high-rewarded trajectories collected from reward-maximizing training. It is noteworthy to emphasize that the symmetric replay does not necessitate additional reward evaluations; *it is free*.

The major benefit of our method is the interoperability of two decomposed training steps: one seeks the high-reward solution, and the other recycles high-rewarded samples to explore symmetric regions for free. Symmetric replay training strategically utilizes previously explored high-rewarded trajectories without additional reward evaluation. The policy is trained to imitate the high-rewarded trajectories in the symmetric space, thereby encouraging the policy to explore under-explored regions. Furthermore, imitating the symmetric trajectories can be regarded as a form of regularization that mitigates overfitting by leveraging symmetric priors during replay training. These symmetric trajectories yield the same solution, yet each trajectory is considered heterogeneous from the policy' perspective. It is advantageous to scale up the replay loop to further utilize collected samples before interacting environment again.

We empirically demonstrate that symmetric replay training consistently improves the sample efficiency by plugging it into various DRL methods with simple implementation. The experimental results show that plugging our method into the state-of-the-art DRL method can achieve superior performances hardware design optimization, and sample efficient molecular optimization by adding it to the competitive DRL method.

## 2 BACKGROUND

Our method aims to improve the sample efficiency of deep reinforcement learning for solving combinatorial optimization with expensive reward function. Specifically, we consider combinatorial optimization as maximization of the (black-box) function $f(\boldsymbol{x})$ over a discrete set $\mathcal{X}$, i.e., $\max_{\boldsymbol{x} \in \mathcal{X}} f(\boldsymbol{x})$. To solve this problem, we formulate the construction and evaluation of the solution as a Markov decision process (MDP). In the MDP, we let each state $s$ describe a subsequence of the action trajectory with problem context $\boldsymbol{c}$, i.e., $s_t = \{(a_1, \ldots, a_{t-1}); \boldsymbol{c}\}$. Therefore, the initial state corresponds to an empty, i.e., $s_1 = \{\emptyset; \boldsymbol{c}\}$, and the final state corresponds to a complete sequence of actions, i.e, $s_T = \{\vec{a}; \boldsymbol{c}\}$, giving a solution $\boldsymbol{x}$. Then, a policy $\pi(a|s)$ decides a transition between states with an action $a$ to update the incomplete solution described by the state $s$. We assume that the transition is deterministic, meaning that the next state is determined by a specific transition function, such as $s_{t+1} \leftarrow s_t \cup \{a_t\}$.

We further impose two conditions on the MDP that exploit the prior knowledge about combinatorial optimization problems considered in this work. First, we assume the reward is episodic, i.e., given a terminated action-state trajectory associated with a solution $\boldsymbol{x}$, the reward $R(s_T) = f(\boldsymbol{x})$ and $R(s_t) = 0$ for $t < T$. Note that the $s_T$ contains the action sequence $\vec{a}$. The next condition is about how the action space $\mathcal{A}_t$ for action $a_t$ made at each state $s_t$ only consists of actions that generate a valid solution for the combinatorial optimization.

## 3 REINFORCEMENT LEARNING WITH SYMMETRIC REPLAY TRAINING

### 3.1 OVERVIEW

Our method, *symmetric replay training (SRT)*, improves the sample efficiency of DRL for CO by exploiting transformed samples without additional reward computation. The key idea is to utilize the existence of solution-preserving symmetric transformation for generating novel action trajectories. Our approach enhances sample efficiency in two-folded: (1) by increasing the number of policy updates per reward calls and (2) by stimulating the exploration of under-explored regions via symmetric trajectories.

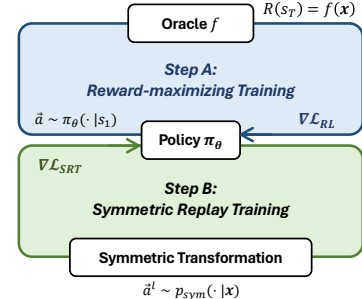

Figure 2: Overview of iterative training

Our method repeats the following two policy update loops:

**Step A: Reward-maximizing training**

Train the (factorized) policy $\pi_\theta(\vec{a}|s_1) = \prod_{t=1}^{T} \pi_\theta(a_t|s_t)$ using a conventional episodic reinforcement learning algorithm.

**Step B: Symmetric replay training**

1. Collect high-rewarded trajectories from Step A.
2. Randomly transform the sampled trajectory using symmetric transformation.
3. Train the policy by imitating the symmetric trajectories.

Intuitively, our **Step A** is designed to encourage the policy to exploit the high-reward samples via reinforcement learning. Then, **Step B** aims to promote exploration of the symmetric space by imitating the collected high-reward samples with symmetric transformation. It is noteworthy that the DRL model and training method in reward-maximizing training are not restricted, allowing the application of various algorithms for episodic tasks as base DRL methods. In the following subsections, we describe the details of a solution-preserving symmetric transformation policy and replay training.

### 3.2 SYMMETRIC TRANSFORMATION POLICY

In this subsection, we provide an explicit characterization of the symmetric transformation used in our algorithm. Symmetric transformation gives another action trajectory that induces the same solution as a given action trajectory, i.e., solution-preserving. To begin with, we introduce a non-injective function $C$ that maps an action trajectory $\vec{a}$ to its corresponding solution $\boldsymbol{x}$ given initial state. It allows for defining symmetry between action trajectories with respect to the solution.

**Definition 1** (Symmetric action trajectories). A pair of action trajectories $\vec{a}^1$ and $\vec{a}^2$ given initial state are symmetric if they induce the same solution $\boldsymbol{x} \in \mathcal{X}$, i.e., if $C(\vec{a}^1) = C(\vec{a}^2) = \boldsymbol{x}$. Also let $\vec{\mathcal{A}}_{\boldsymbol{x}}$ denote a set of symmetric action trajectories for solution $\boldsymbol{x}$, i.e., $\vec{\mathcal{A}}_{\boldsymbol{x}} = \{\vec{a}|C(\vec{a}) = \boldsymbol{x}\}$.

For example, in the traveling salesman problem (TSP), a representative CO problem, cyclically shifting $k$ positions to the left or right gives the same solution. When considering four cities, the sequences 1-2-3-4-1 and 3-4-1-2-3 represent the identical Hamiltonian cycle. Thus, these shifted action sequences are symmetric action trajectories.

**Definition 2** (Symmetric transformation policy). A symmetric transformation policy, denoted by $p_{\mathrm{sym}}(\vec{a}|\boldsymbol{x})$, is a conditional probability distribution over $\vec{\mathcal{A}}_{\boldsymbol{x}}$ for given $\boldsymbol{x}$.

Thus, symmetric trajectories can be sampled from the symmetric transformation policy, i.e., $\vec{a} \sim p_{\text{sym}}(\vec{a}|\boldsymbol{x})$. Here, $\boldsymbol{x}$ is the corresponding solution of an action trajectory sampled from the current training policy, i.e., $\boldsymbol{x} = C(\vec{a})$, where $\vec{a} \sim \pi_\theta(\vec{a}|s_1)$. To maximize symmetric exploration entropy, we set $p_{\text{sym}}(\vec{a}|\boldsymbol{x})$ as uniform distribution; see Theorem 1 in Section 3.4.

## 3.3 SYMMETRIC REPLAY TRAINING

The symmetric replay training (SRT) process involves imitating $L$ symmetrically transformed action trajectories, i.e., $\vec{a}^1, \ldots, \vec{a}^L$. The symmetric replay training loss function is derived as follows:

$$\mathcal{L}_{\text{SRT}} = -\frac{1}{L} \sum_{i=1}^{L} \log \pi_\theta(\vec{a}^i|s_1), \tag{1}$$

The SRT loss function is formulated to maximize the log-likelihood of the symmetric trajectories. To adjust scale of loss function, we introduce a scale coefficient. As a rough guideline, we establish a coefficient that renders the SRT loss approximately 10 to 100 times smaller than $\mathcal{L}_{RL}$ according to the base DRL methods.

Our symmetric replay training significantly benefits from additionally exploring the high-rewarded regions on the symmetric space for free. Moreover, we can explore regions that are likely to have higher rewards but are far from the current regions, since the symmetric action trajectory may have a significant edit distance from the original trajectory.[1]

## 3.4 INTERPLAY OF REWARD-MAXIMIZING AND SYMMETRIC REPLAY TRAINING

This subsection presents an analysis of the interplay between two iterative steps: reward-maximizing training and symmetric replay training. First, we begin with introducing a theorem about the maximization of policy entropy.

**Theorem 1.** Consider a distribution $\pi_\theta(\vec{a}|s_1)$ over the action trajectory from a state $s_1$, and $p(\boldsymbol{x}|s_1)$ over its corresponding solutions. Let $U_{\boldsymbol{x}}(\vec{a}|\boldsymbol{x})$ denote an uniform distribution over $\vec{\mathcal{A}}_{\boldsymbol{x}}$. Then the entropy of $\pi(\vec{a}|s_1)$ can be decomposed and upper bounded as follows:

$$\mathcal{H}(\pi_\theta(\vec{a}|s_1)) = \underbrace{\mathcal{H}(p(\boldsymbol{x}|s_1))}_{\text{Step A}} + \underbrace{\mathbb{E}_{\boldsymbol{x} \sim p(\boldsymbol{x}|s_1)} \mathcal{H}(p_{\text{sym}}(\vec{a}|\boldsymbol{x}))}_{\text{Step B}}$$

$$\leq \mathcal{H}(p(\boldsymbol{x}|s_1)) + \mathbb{E}_{\boldsymbol{x} \sim p(\boldsymbol{x}|s_1)} \mathcal{H}(U_{\boldsymbol{x}}(\vec{a}|\boldsymbol{x})). \tag{2}$$

*Proof.* See Appendix A for the entire proof.

In Theorem 1, the policy entropy $\mathcal{H}(\pi_\theta(\vec{a}|s_1))$ is decomposed into two distinct components: the entropy associated with the solution exploration policy, denoted as $\mathcal{H}(p(\boldsymbol{x}|s_1))$, and the entropy for the symmetric transformation policy, expressed as $\mathbb{E}_{\boldsymbol{x} \sim p(\boldsymbol{x}|s_1)} \mathcal{H}(p_{\text{sym}}(\vec{a}|\boldsymbol{x}))$. This decomposition offers an intuitive illustration of our two-step learning process. In Step A, maximizing entropy encourages the policy to explore high-reward solutions. Subsequently, maximizing the second entropy term encourages the search of various symmetric spaces while preserving high-reward solutions. We achieve this by employing uniform symmetric transformation, represented as $p_{\text{sym}}(\vec{a}|\boldsymbol{x}) = U_{\boldsymbol{x}}(\vec{a}|\boldsymbol{x})$. This approach enables us to maximize entropy exploration within the symmetric space, facilitating a more comprehensive search of potential solutions.

## 4 EXPERIMENTS

This section presents experiments with three distinct settings: a synthetics scenario and two real-world scenarios of hardware design and sample-efficient molecular optimization. For the synthetic setting, we employ traveling salesman problems (TSP), the widely studied CO problems, for precise analyses of the proposed method. We also provide the additional experiment on other synthetic CO problems in Appendix D.4.

---

[1]*Edit distance* is a measure of similarity between two sequences of characters or symbols, defined as the minimum number of operations required to transform one sequence into the other, e.g., *Hamming distance*.

Table 1: Sample efficiency on the synthetic TSP ($N = 50$) with four independent seeds. The average costs ($\downarrow$) and the standard deviations are reported; the improved cost via SRT is denoted in *italics*.

| Method | $K = 100\text{K}$ | $K = 2\text{M}$ |
|---|---|---|
| A2C (Bello et al., 2017) | $6.630 \pm 0.037$ | $6.115 \pm 0.009$ |
| A2C + SRT (ours) | *$6.560 \pm 0.051$* | *$6.037 \pm 0.005$* |
| PG-Rollout (Kool et al., 2018) | $7.138 \pm 0.196$ | $6.226 \pm 0.026$ |
| PG-Rollout + SRT (ours) | *$6.879 \pm 0.110$* | *$6.131 \pm 0.019$* |
| PPO (Schulman et al., 2017) | $6.771 \pm 0.120$ | $6.319 \pm 0.110$ |
| PPO + SRT (ours) | *$6.712 \pm 0.024$* | *$6.249 \pm 0.045$* |
| GFlowNet (Bengio et al., 2021) | $6.880 \pm 0.093$ | $6.203 \pm 0.016$ |
| GFlowNet + SRT (ours) | *$6.621 \pm 0.049$* | *$6.163 \pm 0.008$* |

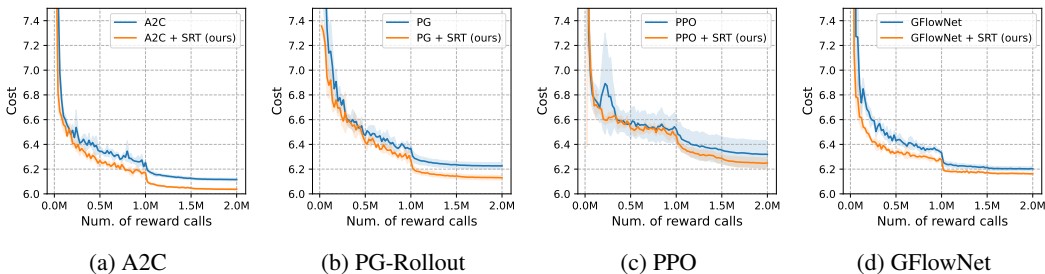

(a) A2C     (b) PG-Rollout     (c) PPO     (d) GFlowNet

Figure 3: Sample efficiency on the synthetic TSP ($N = 50$) with four independent seeds.

In the subsequent sections, we validate the effectiveness of our method in real-world applications. Firstly, we conduct experiments focused on hardware design optimization, particularly addressing the Decap Placement Problem (DPP), a widely recognized problem within the hardware design domain (Koo et al., 2017; Kim et al., 2021; Berto et al., 2023). Furthermore, we extend our experiments to the Practical Molecular Optimization (PMO) benchmark (Gao et al., 2022), a well-established benchmark for sample-efficient molecular optimization.

### 4.1 SYNTHETIC SETTING: TRAVELING SALESMAN PROBLEMS

In general, CO problems have closed forms of objective functions, which means the reward evaluations are not expensive. However, we conduct experiments on the TSP synthetic dataset assuming that the number of computing objective function values is limited; this allows more controlled experiments and more precise analysis.

**Tasks.** Traveling salesman problems (TSP) aim to minimize the distance of a tour that visits all customers and returns to the starting point. In TSP, the distance between consequent customers is defined as an Euclidean distance. The auto-regressive policy starts from an empty tour and constructs the (partial) tour by iteratively selecting the next visit. In TSP, a solution denotes a cycle (i.e., a route) without a designated starting point. Thus, symmetric trajectories are obtained by cyclically shifting $k$ positions to the left or right. Furthermore, in TSP with Euclidean distance, the reversed order of visiting sequence also gives a symmetric action trajectory. We set the maximum reward calls as 2M.

**Experimental setting.** Our experiments are conducted with AM architecture from Kool et al. (2018). We employ various RL methods, including policy gradient with actor-critic (Bello et al., 2017), policy gradient with greedy rollout (Kool et al., 2018), Proximal Policy Optimization (PPO; Schulman et al., 2017), and a Generative Flow Network (GFlowNet; Bengio et al., 2021), and additionally implement SRT on top of the DRL methods to enhance sample efficiency. Note that GFlowNet is an off-policy method. We basically follow the hyperparameter configuration used in AM. For additional parameters of PPO and GFlowNet, we systematically evaluate several combinations to identify the most optimal configuration. The details about implementations are provided in Appendix B. We measure the average costs on the validation dataset over the number of reward calls

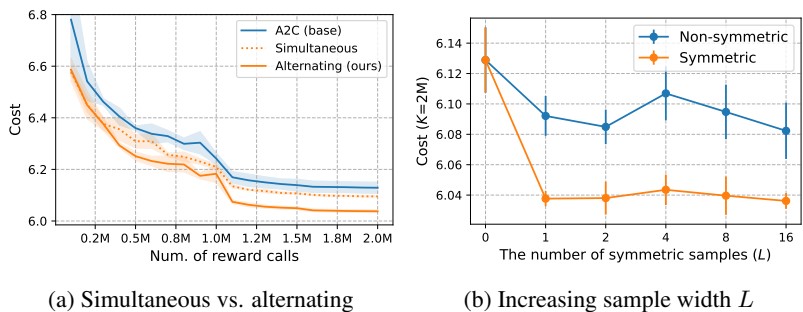

(a) Simultaneous vs. alternating          (b) Increasing sample width $L$

Figure 4: Ablation study for the iterative training steps and sample width in SRT

$(K)$ in training with four independent random seeds. In symmetric replay training, we gather confidence trajectories from the up-to-date policy via greedy rollout, assuming that the greedy solution gives a relatively high reward trajectory.

**Does SRT improve sample efficiency?**  As illustrated in Figure 3 and detailed in Table 1, our approach consistently demonstrates enhanced sample efficiency across various DRL methods. Notably, A2C outperforms the other methods under conditions where the number of available training samples is limited, thereby resulting in the best performance when combined with SRT. The most substantial improvement facilitated by SRT is observed in the case of GFlowNet, where a cost reduction of 3.76% is achieved when the sample size $K = 100$K. In PG-Rollout, a 1.53% cost reduction is realized with SRT when $K = 2$M.

**Ablation study.**  To verity the effectiveness of iterative training steps, we conducted ablation studies using the A2C method on TSP with $N = 50$. In simultaneous update, the policy is updated using the sum of $\mathcal{L}_{RL}$ and $\mathcal{L}_{SRT}$. In detail, after calculating $\mathcal{L}_{RL}$, we collect the high-rewarded samples using greedy rollout without model updating. Then, the model is updated once to minimize total loss. On the other hand, alternating update separates the reward-maximizing training and symmetric replay training. As depicted in Figure 4a, it is observed that alternating Steps A and B more effectively enhances the sample efficiency, even though simultaneous updates also yield improved sample efficiency compared to the original DRL methods. We also provide the results of the ablation study for the loss function in the symmetric replay training and comparative experiment with experience replay and ours in Appendix D.1 and Appendix D.2, respectively.

**Choice of $L$ in SRT.**  We conducted the ablation study for varying sample width in replay training, i.e., $L$ in Eq. (1). 'Non-symmetric' denotes the replay training without symmetric transformation; thus, increased $L$ gives duplicated samples. Figure 4b shows that the symmetric replay training consistently gives better performance and robust to the choice of $L$.

**Overfitting in replay training.**  In this section, we investigate the extent to which symmetric replay training can increase the number of replay loops without encountering issues of overfitting. The experiments are conducted by increasing the replay training loops with the fixed sample width ($L = 1$). To assess the effectiveness of symmetric transformation in replay training, we perform the same experiments without symmetric transformation (the results are denoted as 'Non-symmetric' in Figure 5). As illustrated in Figure 5, our symmetric replay training successfully enhances sample efficiency up to replay

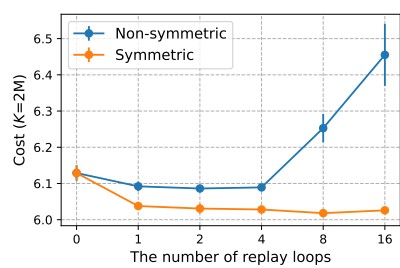

Figure 5: Increasing replay loop

loops 16. In contrast, non-symmetric replay training experiences diminished performance when reaching when the loop exceeds 8. This suggests that the symmetric transformation provides trajectories that induce the same solution but are distinctive from the policy's perspective, contributing to mitigating overfitting from replaying the restricted set of repetitive trajectories.

Table 2: Experimental results of DPP on two different PDN environments. All experiments are done in four times; the average rewards (↑) and standard deviations are reported.

| Method | Chip-package PDN | HBM PDN |
|---|---|---|
| A2C (Kool et al., 2018) | 9.772 ± 0.823 | 25.945 ± 0.177 |
| A2C + SRT (ours) | *12.757 ± 0.267* | *26.449 ± 0.094* |
| PG-Rollout (Kool et al., 2018) | 10.240 ± 0.955 | 25.714 ± 0.122 |
| PG-Rollout + SRT (ours) | *12.601 ± 0.467* | *26.355 ± 0.013* |
| PPO (Schulman et al., 2017) | 9.821 ± 0.411 | 25.907 ± 0.068 |
| PPO + SRT (ours) | *11.279 ± 0.511* | *26.322 ± 0.141* |
| GFlowNet (Bengio et al., 2021) | 10.403 ± 0.411 | 25.930 ± 0.078 |
| GFlowNet + SRT (ours) | *12.772 ± 0.276* | *26.316 ± 0.090* |

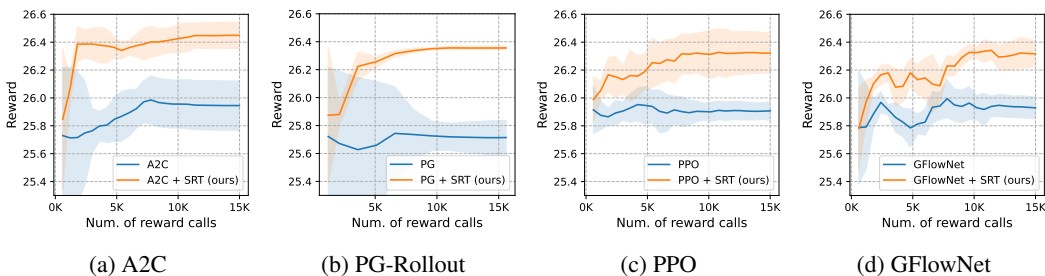

(a) A2C     (b) PG-Rollout     (c) PPO     (d) GFlowNet

Figure 6: Sample efficiency on HBM PDN with for independent seeds.

## 4.2 HARDWARE DESIGN OPTIMIZATION: DECAP PLACEMENT PROBLEMS

**Tasks.** We employ decoupling capacitor placement problems (DPP), which constitute a fundamental optimization challenge in hardware design. In the context of hardware devices like CPUs and GPUs, a decoupling capacitor (decap) is a critical component responsible for reducing power noises along the power distribution network (PDN). The primary objective of DPP is to identify the optimal arrangement for placing these decaps to maximize the power integrity (PI) objective, which involves computationally expensive evaluations. We tackle two distinct DPP tasks: the chip-package PDN (Koo et al., 2017) and the High Bandwidth Memory (HBM) PDN (Jun et al., 2017) with 15K limited reward calls. These two tasks exhibit variations in their PI landscapes, presenting unique challenges and considerations. In DPP, permuting the decision orders of decap yields symmetric trajectories.

**Experimental setting.** We employ a Device Transformer (DevFormer; Kim et al., 2023) as our base neural architecture, originally designed for offline DPP tasks. In an online optimization context involving interactions with the DPP environment, we integrate well-established DRL methods (A2C, PG, PPO, and GFlowNet) on DevFormer to maximize effectiveness. Similar to the experiment on TSP, we adjust hyperparameters by examining several combinations since DPP tasks have different reward scales. The details are provided in Appendix B. To verify the effectiveness of our methods, SRT is built on these DRL methods to enhance sample efficiency and compare the resulting rewards.

**Results.** Addressing DPP tasks proved challenging due to the limited online reward calls available. For instance, the PG-Rollout method exhibited high variances, making it less effective at exploring the DPP solution space. This challenge stems from the inherent symmetry of DPP, where multiple trajectories could lead to identical solutions. Conversely, GFlowNet models, incorporating structured bias to handle solution symmetries, showed improved performance compared to non-symmetric DRL methods like PPO. As demonstrated in Table 2 and Figure 6, significant improvements are observed by applying SRT on top of these methods. The most considerable reward improvement is observed in A2C for chip-package PDN, at 30.54%. In the case of GFlowNet with SRT, exploring focused on the symmetric variants within the high-reward region enhances the sample efficiency by mitigating its underfitting.

Table 3: Experimental results on sample efficient molecular optimization. Area under the curve of top-10 average property value (↑) is reported. **Bold** indicates the best performance among the presented methods in this table, while *italic* denotes an enhanced performance via SRT.

| Oracle | MolDQN | Model-based GFlowNet | GFlowNet | REINVENT | GFlowNet + SRT (ours) | REINVENT + SRT (ours) |
|---|---|---|---|---|---|---|
| albuterol_similarity | $0.322 \pm 0.009$ | $0.382 \pm 0.010$ | $0.459 \pm 0.028$ | $0.849 \pm 0.021$ | *0.526 ± 0.022* | ***0.894 ± 0.013*** |
| amlodipine_mpo | $0.316 \pm 0.012$ | $0.428 \pm 0.002$ | $0.437 \pm 0.007$ | $0.604 \pm 0.012$ | *0.448 ± 0.010* | ***0.618 ± 0.011*** |
| celecoxib_rediscovery | $0.099 \pm 0.008$ | $0.263 \pm 0.009$ | $0.326 \pm 0.008$ | **0.604 ± 0.087** | *0.345 ± 0.011* | $0.586 \pm 0.071$ |
| deco_hop | $0.551 \pm 0.002$ | $0.534 \pm 0.096$ | $0.587 \pm 0.002$ | $0.629 \pm 0.009$ | $0.582 \pm 0.004$ | ***0.635 ± 0.009*** |
| drd2 | $0.027 \pm 0.002$ | $0.480 \pm 0.075$ | $0.601 \pm 0.055$ | $0.953 \pm 0.006$ | *0.796 ± 0.054* | ***0.960 ± 0.005*** |
| fexofenadine_mpo | $0.483 \pm 0.008$ | $0.689 \pm 0.003$ | $0.700 \pm 0.005$ | $0.736 \pm 0.003$ | $0.688 \pm 0.006$ | ***0.760 ± 0.010*** |
| gsk3b | $0.242 \pm 0.008$ | $0.589 \pm 0.009$ | $0.666 \pm 0.006$ | $0.801 \pm 0.013$ | $0.657 \pm 0.010$ | ***0.837 ± 0.039*** |
| isomers_c7h8n2o2 | $0.430 \pm 0.037$ | $0.791 \pm 0.024$ | $0.468 \pm 0.211$ | $0.887 \pm 0.026$ | *0.928 ± 0.006* | ***0.945 ± 0.012*** |
| isomers_c9h10n2o2pf2cl | $0.331 \pm 0.037$ | $0.576 \pm 0.021$ | $0.199 \pm 0.199$ | $0.753 \pm 0.044$ | *0.628 ± 0.024* | ***0.858 ± 0.027*** |
| jnk3 | $0.099 \pm 0.005$ | $0.359 \pm 0.009$ | $0.442 \pm 0.017$ | $0.589 \pm 0.093$ | *0.505 ± 0.040* | ***0.691 ± 0.066*** |
| median1 | $0.123 \pm 0.006$ | $0.192 \pm 0.003$ | $0.207 \pm 0.003$ | $0.348 \pm 0.009$ | *0.211 ± 0.002* | ***0.362 ± 0.014*** |
| median2 | $0.087 \pm 0.005$ | $0.174 \pm 0.002$ | $0.181 \pm 0.002$ | $0.255 \pm 0.011$ | $0.181 \pm 0.002$ | ***0.256 ± 0.005*** |
| mestranol_similarity | $0.185 \pm 0.024$ | $0.291 \pm 0.005$ | $0.332 \pm 0.012$ | **0.639 ± 0.021** | *0.339 ± 0.004* | $0.629 \pm 0.023$ |
| osimertinib_mpo | $0.672 \pm 0.012$ | $0.787 \pm 0.002$ | $0.785 \pm 0.003$ | $0.820 \pm 0.008$ | $0.784 \pm 0.003$ | ***0.826 ± 0.008*** |
| perindopril_mpo | $0.225 \pm 0.024$ | $0.423 \pm 0.006$ | $0.434 \pm 0.006$ | $0.525 \pm 0.017$ | $0.429 \pm 0.010$ | ***0.540 ± 0.026*** |
| qed | $0.732 \pm 0.017$ | $0.904 \pm 0.002$ | $0.917 \pm 0.002$ | **0.940 ± 0.001** | *0.922 ± 0.002* | **0.940 ± 0.000** |
| ranolazine_mpo | $0.037 \pm 0.010$ | $0.626 \pm 0.005$ | $0.660 \pm 0.004$ | $0.750 \pm 0.013$ | $0.652 \pm 0.008$ | ***0.773 ± 0.019*** |
| scaffold_hop | $0.407 \pm 0.009$ | $0.461 \pm 0.002$ | $0.464 \pm 0.003$ | $0.526 \pm 0.019$ | $0.466 \pm 0.003$ | ***0.534 ± 0.017*** |
| sitagliptin_mpo | $0.040 \pm 0.017$ | $0.180 \pm 0.012$ | $0.217 \pm 0.022$ | $0.481 \pm 0.041$ | *0.282 ± 0.013* | ***0.495 ± 0.061*** |
| thiothixene_rediscovery | $0.101 \pm 0.007$ | $0.261 \pm 0.004$ | $0.292 \pm 0.009$ | $0.485 \pm 0.022$ | $0.291 \pm 0.007$ | ***0.513 ± 0.028*** |
| troglitazone_rediscovery | $0.125 \pm 0.005$ | $0.183 \pm 0.001$ | $0.190 \pm 0.002$ | $0.342 \pm 0.017$ | $0.189 \pm 0.004$ | ***0.358 ± 0.016*** |
| valsartan_smarts | **0.000 ± 0.000** | **0.000 ± 0.000** | **0.000 ± 0.000** | **0.000 ± 0.000** | **0.000 ± 0.000** | **0.000 ± 0.000** |
| zaleplon_mpo | $0.057 \pm 0.028$ | $0.308 \pm 0.027$ | $0.353 \pm 0.024$ | $0.506 \pm 0.010$ | *0.398 ± 0.010* | ***0.522 ± 0.022*** |
| **Average** | 0.247 | 0.430 | 0.431 | 0.610 | *0.489* | ***0.632*** |
| **Num. of 1st Place** | 1/23 | 1/23 | 1/23 | 4/23 | 1/23 | **21/23** |

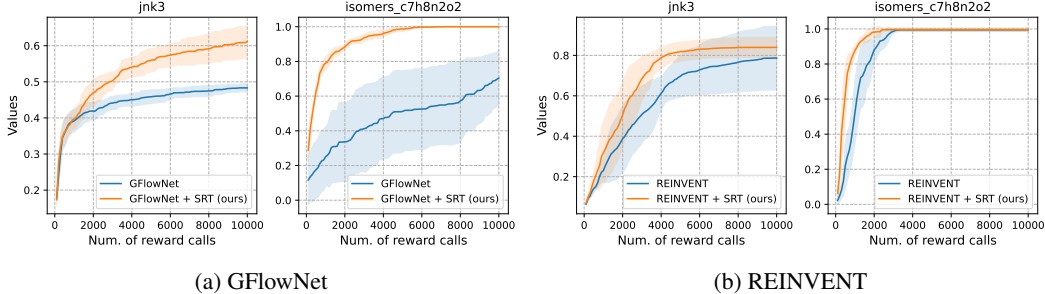

(a) GFlowNet      (b) REINVENT

Figure 7: Average reward of top-10 in `jnk3` and `isomers_c7h8n2o2`.

### 4.3 MOLECULAR OPTIMIZATION: PRACTICAL MOLECULAR OPTIMIZATION BENCHMARK

**Tasks.** We employ practical molecular optimization (PMO; Gao et al., 2022), whose reward evaluations are limited up to 10K, and the goal is achieving the highest score within the limited Oracle calls. PMO contains 23 tasks based on different score functions called Oracles; a task is a CO problem that maximizes the given score function, such as QED (Bickerton et al., 2012), DRD2 (Olivecrona et al., 2017) and JNK3 (Li et al., 2018). For example, QED measures drug safety, while DRD2 and JNK3 measure bioactivities against their corresponding disease targets. In de novo molecular optimization, molecules are represented as graphs or strings,[2] which have multiple ways to describe the same molecule.

**Experimental setting.** We build SRT on the GFlowNet (Bengio et al., 2021) and REINVENT (Olivecrona et al., 2017). Note that REINVENT contains the online experience buffer, so we also collect the replay samples from the experience buffer with the same replay size. SRT is compared to not only its base DRL methods, but also MolDQN (Zhou et al., 2019) and model-based GFlowNet (Jain et al., 2022). It is noteworthy that REINVENT is considered state-of-the-art (SOTA) in the

---

[2] In this study, we employ SELFreferencIng Embedded Strings (SELFIES; Krenn et al., 2020) for the string -based molecular representation to ensure compliance with chemical constraints such as the octet rule.

PMO benchmark. The performance of the methods is evaluated based on the area under the curve (AUC) to consider a combination of optimization ability and sample efficiency. The AUC of the top 10 average performance is mainly reported since it is essential to find distinct molecular candidates for the next stage in drug discovery. Every experiment is conducted with five independent seeds.

**Results.** As shown in Table 3, SRT outperforms 21 out of 23 tasks, attaining an average AUC-10 score of 0.632, which exceeds that of the SOTA method, REINVENT, with an average score of 0.610. In addition, our method verifies its effectiveness by achieving a 3.61% score improvement in REINVENT and 13.46% in GFlowNet on average. Figure 7 further demonstrates our method's effectiveness by illustrating the average scores of the top-10 molecules in the JNK3 and isomer tasks, revealing a substantial enhancement in sample efficiency.

## 5 RELATED WORKS

### 5.1 SYMMETRIES IN DEEP REINFORCEMENT LEARNING FOR CO

Building on the success of the Attention Model (AM; Kool et al., 2018), the Policy Optimization for Multiple Optima (POMO; Kwon et al., 2020) and Symmetric Neural Combinatorial Optimization (Sym-NCO; Kim et al., 2022b) were suggested. They introduce an effective REINFORCE baseline by leveraging symmetries in CO. However, they becomes computationally infeasible in the context of black-box CO. Separately, a Generative Flow Network (GFlowNet; Bengio et al., 2021), which employs a directed acyclic graph (DAG) to represent the combinatorial space in CO problems, was proposed. On DAG structure, each state and a policy correspond to a node and its flow, and the policy is trained to match flow equations, leading solution symmetries. In Malkin et al. (2022) where the trajectory-balance loss is introduced, the backward policy $P_B(-|s)$ is set to be uniform over all the parents of a state $s$. Our uniformly randomized symmetric transformation policy has a similar role to the uniform $P_B$. We provide further related works in Appendix E.1.

Equivariant DRL has also been extensively studied in recent years Mondal et al. (2022); Van der Pol et al. (2020); Mondal et al. (2020); Wang & Walters (2022); Deac et al. (2023). This approach reduces search space by cutting out symmetric space using equivariant representation learning, such as employing equivariant neural networks Cohen & Welling (2016); Weiler & Cesa (2019); Satorras et al. (2021). Consequently, it leads to better generalization and sample efficiency. Being different from these approaches, we focus on handling symmetries in decision space by exploring the symmetric space without restrictions on network structure. Therefore, employing equivariant DRL methods with our method is available when guaranteeing equivariance is crucial.

### 5.2 REPLAY RATIO SCALING

Increasing the number of replay loops is highly related with scaling up replay ratio, which means the number of parameter updates per environment interaction (Wang et al., 2016; Fedus et al., 2020; D'Oro et al., 2022). Replay ratio is also known as update-to-data (UTD) ratio (Chen & Tian, 2019; Smith et al., 2022). Although the benefits of replay ratio scaling are limited, it has demonstrated improved performance, particularly on well-tuned algorithms (Kielak, 2019; Chen & Tian, 2019; Smith et al., 2022). Recently, D'Oro et al. (2022) achieved better replay ratio scaling with parameter reset strategy by mitigating the loss of ability to generalize on model-free RL.

We also provide the related works for black-box combinatorial optimization in Appendix E.2.

## 6 CONCLUSION

This study proposes a new approach, called *symmetric replay training (SRT)*, to enhance the sample efficiency of DRL methods for black-box combinatorial optimization problems. Our approach improves sample by reusing the high-rewarded samples from the policy in the symmetric space, which helps with exploring new regions without additional reward computation. Replay training through symmetric transformations enhances the sample efficiency by effectively increasing the replay ratio while mitigating the adverse effects of overfitting. The experimental results demonstrate the enhanced sample efficiency of the proposed method on various DRL methods in the real-world benchmark, such as hardware design and de novo molecular optimization.

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

# A   PROOF FOR THEOREM 1

Consider $\pi_\theta(\vec{a}|s_1)$ as the distribution over the action trajectory from a state $s_1$ which describes the problem context. Let $\vec{\mathcal{A}}_{\boldsymbol{x}}$ denote the space of action-trajectories associated with the solution $\boldsymbol{x}$.

$$
\begin{aligned}
\mathcal{H}(\pi_\theta(\vec{a}|s_1)) &= -\sum_{\vec{a}\in\vec{\mathcal{A}}} \pi_\theta(\vec{a}|s_1) \log \pi_\theta(\vec{a}|s_1) \\
&= -\sum_{\boldsymbol{x}\in\mathcal{X}} \sum_{\vec{a}\in\vec{\mathcal{A}}_{\boldsymbol{x}}} \pi_\theta(\vec{a}|s_1) \log \pi_\theta(\vec{a}|s_1) \\
&= -\sum_{\boldsymbol{x}\in\mathcal{X}} \sum_{\vec{a}\in\vec{\mathcal{A}}_{\boldsymbol{x}}} p_{\text{sym}}(\vec{a}|\boldsymbol{x})p(\boldsymbol{x}|s_1) \left(\log p_{\text{sym}}(\vec{a}|\boldsymbol{x}) + \log p(\boldsymbol{x}|s_1)\right) \\
&= \mathcal{H}(p(\boldsymbol{x}|s_1)) + \mathbb{E}_{\boldsymbol{x}\sim p(\boldsymbol{x}|s_1)}\mathcal{H}(p_{\text{sym}}(\vec{a}|\boldsymbol{x})) \\
&\le \mathcal{H}(p(\boldsymbol{x}|s_1)) + \mathbb{E}_{\boldsymbol{x}\sim p(\boldsymbol{x}|s_1)}\mathcal{H}(U_{\boldsymbol{x}}(\vec{a}|\boldsymbol{x})),
\end{aligned}
$$

where $U_{\boldsymbol{x}}(\vec{a}|\boldsymbol{x})$ is a uniform distribution over action-trajectories associated with the solution $\boldsymbol{x}$. The third equality stems from the fact that $\pi_\theta(\vec{a}|s_1) = \pi_\theta(\vec{a}, \boldsymbol{x}|s_1)$ since $\boldsymbol{x}$ is fixed given $\vec{a}$. One can show that the final upper-bound is the entropy of distribution obtained from replacing $p_{\text{sym}}(\vec{a}|\boldsymbol{x})$ by $U_{\boldsymbol{x}}(\vec{a}|\boldsymbol{x})$.

# B  IMPLEMENTATION DETAILS

## B.1  PSEUDO-CODE FOR REINFORCE WITH SRT

In this section, we provide a pseudo-code in case of employing REINFORCE as a base DRL method in Step A.

---

**Algorithm 1** REINFORCE with SRT

---

**Input:** The maximum number of reward calls $K$, batch size $B$, SRT sample width $L$, a scale coefficient $\alpha$
**Output:** The trained policy $\pi_\theta$

1: Initialize parameter $\theta$
2: Initialize the number of reward calls $k \leftarrow 0$
3: **while** $k \leq K$ **do**
4:     $s_1^i \leftarrow$ GetInitialState(), $\quad \forall i \in \{1, \ldots, B\}$
5:     $\vec{a} \sim \pi_\theta(\vec{a}|s_1) = \prod_{t=1}^{T} \pi_\theta(a_t|s_t)$
6:     $\triangleright$ REWARD EVALUATION
7:     Evaluate $R(\boldsymbol{x}^i) = R(C(\vec{a}^i)), \quad \forall i \in \{1, \ldots, B\}$
8:     $k \leftarrow k + B$
9:     $\triangleright$ STEP A. REWARD-MAXIMIZING TRAINING
10:    $\nabla \mathcal{L}_{RL} \leftarrow \sum_{i=1}^{B}(R(\boldsymbol{x}^i) - R_{BL})\nabla \log \pi_\theta(\cdot)$ $\qquad\qquad \triangleright R_{BL}$ *is a REINFORCE baseline*
11:    $\theta \leftarrow \text{Adam}(\theta, \nabla \mathcal{L}_{RL})$
12:    $\triangleright$ STEP B. SYMMETRIC REPLAY TRAINING
13:    $\vec{a}^i \leftarrow$ GreedyRollout(), $\quad \forall i \in \{1, \ldots, B\}$
14:    $\vec{a}^{i,1}, \ldots, \vec{a}^{i,L} \sim p_{sym}(\cdot|\boldsymbol{x}^i), \quad \forall i \in \{1, \ldots, B\}$
15:    $\nabla \mathcal{L}_{\text{SRT}} \leftarrow \alpha \sum_{i=1}^{B} \frac{1}{L} \sum_{l=1}^{L} \nabla \log \pi_\theta(\vec{a}^{i,l}|s_1^i)$
16:    $\theta \leftarrow \text{Adam}(\theta, \nabla \mathcal{L}_{\text{SRT}})$
17: **end while**

---

There are various ways to collect high-reward samples in Step B. For example, GreedyRollout can be replaced with other strategies, such as reward-prioritized sampling or collecting Top-$k$ samples.

## B.2  PROXIMAL POLICY OPTIMIZATION (PPO)

We use AM architecture (Kool et al., 2018) on TSP and Devformer architecture (Kim et al., 2023) on DPP for parameterizing compositional policy $\pi(\boldsymbol{x}|s_1) = \prod_{t=1}^{N} \pi(a_t|s_t)$. Then, we implement based on the following equation as follows:

$$\mathcal{L}(\boldsymbol{x}; s_1) = \min\left[ A(\boldsymbol{x}; s_1)\frac{\pi(\boldsymbol{x}|s_1)}{\pi_{\text{old}}(\boldsymbol{x}|s_1)}, A(\boldsymbol{x}; s_1)\text{clip}\left( \frac{\pi(\boldsymbol{x}|s_1)}{\pi_{\text{old}}(\boldsymbol{x}|s_1)}, 1 - \epsilon, 1 + \epsilon \right) \right],$$

$$A(\boldsymbol{x}; s_1) = R(\boldsymbol{x}; s_1) - V(s_1),$$

where $R$ stands for reward function and $V$ stands for value function. Since we implement PPO on compositional MDP setting, we train value function in the context of $s_1$ by following actor-critic implementation of Kool et al. (2018).

**Hyperparameters.**  We systematically investigate a range of hyperparameter combinations involving different baselines ([rollout, critic]), various values for clipping epsilon ([0.1, 0.2, 0.3]), and numbers of inner loops ([5, 10, 20]). Our observations reveal that the critic baseline consistently enhances training stability across all tasks, leading to reduced variation when modifying the training seeds. The best configurations for each task are provided in Table 4.

## B.3  GENERATIVE FLOW NETWORK (GFLOWNET)

Similar to the PPO implementation, we employ the Attention Model (AM) architecture (Kool et al., 2018) on the Traveling Salesman Problem (TSP), and the DevFormer architecture (Kim et al., 2023) on DPP, for parameterizing the compositional forward policy $P_F(\tau|s_1) = \prod_{t=1}^{N} P_F(a_t|s_t)$. Subsequently, we configure the backward policy $P_B$ as a uniform distribution for all possible parent nodes,

Table 4: Hyperparameter configurations for PPO.

|  | TSP | Chip-package PDN | HBM PDN |
|---|---|---|---|
| Baseline | critic | critic | critic |
| Eps. clip | 0.2 | 0.1 | 0.2 |
| Number of inner loops $k$ | 5 | 20 | 10 |

following the methodology outlined in (Malkin et al., 2022). Lastly, we parameterize $Z(s_1)$ using a two-layer perceptron with ReLU activation functions, where the number of hidden units matches the embedding dimension of the AM or DevFormer. This two-layer perceptron takes input from the mean of the encoded embedding vector obtained from the encoder of the AM or DevFormer and produces a scalar value to estimate the partition function.

To train the GFlowNet model, we use trajectory balance loss introduced in Malkin et al. (2022) as follows:

$$\mathcal{L}(\tau; s_1) = \left( \log \left( \frac{Z(s_1) P_F(\tau|s_1)}{e^{-\beta E(\boldsymbol{x}; s_1)} P_B(\tau|s_1)} \right) \right)^2 \tag{3}$$

The trajectory $\tau$ includes a terminal state represented as $\boldsymbol{x}$. Subsequently, we employ an on-policy optimization method to minimize Eq. (3), with trajectories $\tau$ sampled from the training policy $P_F$. In this context, $E(\boldsymbol{x}; s_1)$ represents the energy, which is essentially the negative counterpart of the reward $R(\boldsymbol{x}; s_1)$. The hyperparameter $\beta$ plays the role of temperature adjustment in this process.

**Hyperparameters.** We explore a spectrum of hyperparameter combinations, varying $\beta$ ([5, 10, 20]) and numbers of inner loops ([2, 5, 10]). The best configurations for each task are provided in Table 5.

Table 5: Hyperparameter configurations for GFlowNet.

|  | TSP | Chip-package PDN | HBM PDN |
|---|---|---|---|
| $\beta$ | 20 | 10 | 10 |
| Number of inner loops $k$ | 10 | 2 | 2 |

## C  EXPERIMENTAL DETAILS

### C.1  TRAVELING SALESMAN PROBLEMS (TSP)

Since we employ the AM architecture, we use the same hyperparameters for the model architecture and training parameters except for the batch and epoch data sizes.[3]  Initially, the Attention Model (AM) employed a batch size of 512 and an epoch data size of 1,280,000. Notably, the evaluation of the greedy rollout baseline was conducted every epoch. When the number of available training samples is constrained, utilizing a smaller batch size and epoch data size becomes advantageous. Consequently, we adjusted these parameters to be 100 for batch size and 10,000 for epoch data size. In symmetric replay training (Step B), the scale coefficients are meticulously set to scale the SRT loss. As a rough guideline, we establish a coefficient that renders the SRT loss approximately 10 to 100 times smaller than the RL loss. Additionally, for the number of symmetric transformations ($L$ in Eq. (1)) is set as the number of inner loops. See Table 6 in details.

Table 6: Scale coefficient and the number of symmetric transformations in TSP.

|                   | A2C   | PG-Rollout | PPO      | GFlowNet   |
|-------------------|-------|------------|----------|------------|
| Scale coefficient | 0.001 | 0.001      | 0.00001  | 0.1        |
| $L$               | 1     | 1          | $5 (= k)$ | $10 (= k)$ |

### C.2  DECAP PLACEMENT PROBLEMS (DPP)

Similar to the experiments on TSP, we follow the setting of DevFormer.[4]  We set the batch size as 100 and epoch data size as 600. Note that the maximum number of reward calls is set 15K,, a considerably smaller limit compared to TSP. Regarding the scale coefficient and the number of symmetric transformations, we maintain consistency with the principles applied in the TSP experiments as follows:

Table 7: Scale coefficient and the number of symmetric transformations in DPP tasks.

|                   | A2C  | PG-Rollout | PPO       | GFlowNet  |
|-------------------|------|------------|-----------|-----------|
| Scale coefficient | 0.01 | 0.01       | 0.01      | 0.1       |
| $L$               | 1    | 1          | $20 (= k)$ | $2 (= k)$ |

### C.3  PRACTICAL MOLECULAR OPTIMIZATION (PMO)

We basically follow the experimental setting (e.g., batch size) in the practical molecular optimization (PMO) benchmark.[5]  In symmetric replay training, we utilize reward-prioritized sampling for the online buffer, which contains molecules generated during online learning. For the REINVENT, where the replay buffer is already incorporated, we set the number of replaying samples equal to the replay buffer size, i.e., 24, and the scale coefficient to 0.001. Regarding the GFlowNet method, we configure the number of replaying samples to match the batch size of 64. Furthermore, we set the scale coefficient to 1.0, given that the RL loss in GFlowNet is much higher compared to REINVENT.

---

[3]AM: `https://github.com/wouterkool/attention-learn-to-route`
[4]DevFormer: `https://github.com/kaist-silab/devformer`
[5]Practical molecular optimization: `https://github.com/wenhao-gao/mol_opt`

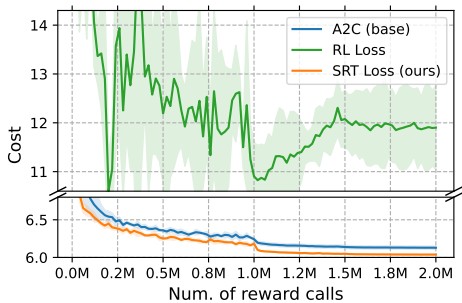

Figure 8: Ablation study for the loss function. The average validation costs over computation budget are measured on TSP50.

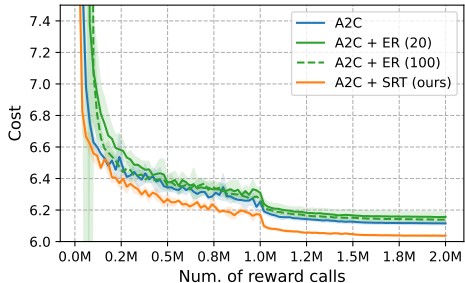

Figure 9: Comparative experiments with experience replay and SRT. The average validation costs over computation budget are measured on TSP50.

## D ADDITIONAL EXPERIMENTS

### D.1 ABLATION STUDY FOR LOSS FUNCTIONS ON TSP50

This subsection provide the results of ablation study for using imitation loss in symmetric replay training. The experiments are conducted with the same RL loss (denoted as 'RL Loss') and the proposed imitation loss (denoted as 'SRT Loss'). Since we employ the A2C as the base, RL loss is as follows:

$$\mathcal{L}_{RL} = \frac{1}{B} \sum_{i=1}^{B} \left(R(\boldsymbol{x}|s_1) - V(s_1)\right) \log \pi_\theta(\vec{a}|s_1),$$

where $B$ is batch size. As shown in Figure 8, it is evident that symmetric replay training with RL loss exhibits instability. This is a natural consequence since the symmetric trajectories often diverge significantly from the current policy.

### D.2 COMPARISON WITH EXPERIENCE REPLAY ON TSP50

We conduct comparative experiments with experience replay and ours using A2C on TSP50. This requires importance sampling weight as follows:

$$\mathbb{E}_{\vec{a}\sim p(\cdot|s_1)}[R(\boldsymbol{x})] = E_{\vec{a}\sim q(\cdot|s_1)}\left[\frac{p(\vec{a}|s_1)}{q(\vec{a}|s_1)}R(\boldsymbol{x})\right] = \mathbb{E}_{\vec{a}\sim q(\cdot|s_1)}\left[\prod_{t=1}^{T}\frac{p(a_t|s_t)}{q(a_t|s_t)}R(\boldsymbol{x})\right],$$

where $p(\cdot|s_1)$ and $q(\cdot|s_1)$ are the training policy and behavior policy, respectively, and $\boldsymbol{x} = C(\vec{a})$.

The results show that existing experience replay method can suffer from high-variance because of importance sampling, leading to the degradation of performance. On the other hand, ours does not require the importance sampling weight since SRT is imitation learning.

Table 8: Experimental results on sample efficient molecular optimization. Area under the curve of top-10 average property value (↑) is reported with *three* independent seeds.

| Oracle | GPBO | Graph GA | REINVENT | REINVENT + SRT (ours) |
|---|---|---|---|---|
| albuterol_similarity | $0.896 \pm 0.009$ | $0.838 \pm 0.027$ | $0.847 \pm 0.018$ | $0.898 \pm 0.015$ |
| amlodipine_mpo | $0.577 \pm 0.042$ | $0.649 \pm 0.014$ | $0.603 \pm 0.015$ | $0.623 \pm 0.006$ |
| celecoxib_rediscovery | $0.733 \pm 0.026$ | $0.682 \pm 0.127$ | $0.573 \pm 0.062$ | $0.605 \pm 0.086$ |
| deco_hop | $0.620 \pm 0.008$ | $0.601 \pm 0.004$ | $0.624 \pm 0.006$ | $0.636 \pm 0.009$ |
| drd2 | $0.933 \pm 0.014$ | $0.968 \pm 0.006$ | $0.957 \pm 0.005$ | $0.961 \pm 0.007$ |
| fexofenadine_mpo | $0.723 \pm 0.002$ | $0.773 \pm 0.014$ | $0.736 \pm 0.001$ | $0.762 \pm 0.012$ |
| gsk3b | $0.878 \pm 0.018$ | $0.792 \pm 0.092$ | $0.802 \pm 0.016$ | $0.818 \pm 0.040$ |
| isomers_c7h8n2o2 | $0.912 \pm 0.023$ | $0.944 \pm 0.030$ | $0.871 \pm 0.022$ | $0.940 \pm 0.013$ |
| isomers_c9h10n2o2pf2cl | $0.542 \pm 0.383$ | $0.831 \pm 0.018$ | $0.779 \pm 0.015$ | $0.854 \pm 0.026$ |
| jnk3 | $0.588 \pm 0.095$ | $0.677 \pm 0.120$ | $0.649 \pm 0.027$ | $0.710 \pm 0.046$ |
| median1 | $0.288 \pm 0.003$ | $0.265 \pm 0.016$ | $0.345 \pm 0.010$ | $0.363 \pm 0.017$ |
| median2 | $0.298 \pm 0.005$ | $0.268 \pm 0.013$ | $0.257 \pm 0.010$ | $0.252 \pm 0.002$ |
| mestranol_similarity | $0.659 \pm 0.108$ | $0.550 \pm 0.032$ | $0.627 \pm 0.018$ | $0.642 \pm 0.019$ |
| osimertinib_mpo | $0.788 \pm 0.003$ | $0.818 \pm 0.007$ | $0.818 \pm 0.004$ | $0.830 \pm 0.009$ |
| perindopril_mpo | $0.495 \pm 0.005$ | $0.498 \pm 0.009$ | $0.526 \pm 0.021$ | $0.553 \pm 0.027$ |
| qed | $0.936 \pm 0.001$ | $0.939 \pm 0.001$ | $0.940 \pm 0.001$ | $0.940 \pm 0.000$ |
| ranolazine_mpo | $0.737 \pm 0.007$ | $0.716 \pm 0.011$ | $0.756 \pm 0.014$ | $0.777 \pm 0.024$ |
| scaffold_hop | $0.536 \pm 0.007$ | $0.506 \pm 0.016$ | $0.519 \pm 0.021$ | $0.533 \pm 0.017$ |
| sitagliptin_mpo | $0.422 \pm 0.008$ | $0.486 \pm 0.007$ | $0.512 \pm 0.018$ | $0.498 \pm 0.053$ |
| thiothixene_rediscovery | $0.565 \pm 0.032$ | $0.494 \pm 0.010$ | $0.489 \pm 0.026$ | $0.502 \pm 0.030$ |
| troglitazone_rediscovery | $0.417 \pm 0.025$ | $0.421 \pm 0.041$ | $0.356 \pm 0.002$ | $0.346 \pm 0.007$ |
| valsartan_smarts | $0.000 \pm 0.000$ | $0.000 \pm 0.000$ | $0.000 \pm 0.000$ | $0.000 \pm 0.000$ |
| zaleplon_mpo | $0.456 \pm 0.019$ | $0.449 \pm 0.012$ | $0.503 \pm 0.012$ | $0.509 \pm 0.016$ |
| **Average** | 0.609 | 0.616 | 0.613 | 0.633 |

### D.3 ADDITIONAL RESULTS ON MOLECULAR OPTIMIZATION

In this subsection, we provide the additional results of black-box optimization method in the PMO benchmark. Gaussian process Bayesian optimization (GPBO; Tripp et al., 2021) and Graph Genetic Algorithm (Graph GA; Jensen, 2019) are included. Note that GPBO employs Graph GA when optimizing the GP acquisition function. Though Graph GA demonstrates powerful performance, designing operators, such as crossover and mutation, greatly affects performance (Li et al., 2022) Thus, the careful algorithm design, which requires specific domain knowledge, is necessitated whenever there is a change in tasks. The results show that SRT outperforms other black-box optimization method by improving on-policy RL method, REINVENT.

Table 9: Experimental results on sample efficient Euclidean CO problems.

| | Method | $N = 50$ | | $N = 100$ | |
|---|---|---|---|---|---|
| | | $K = 200K$ | $K = 2M$ | $K = 200K$ | $K = 2M$ |
| TSP | AM Critic | $6.541 \pm 0.075$ | $6.129 \pm 0.021$ | $9.600 \pm 0.090$ | $8.917 \pm 0.115$ |
| | AM Rollout | $6.708 \pm 0.077$ | $6.199 \pm 0.014$ | $11.891 \pm 1.008$ | $9.193 \pm 0.053$ |
| | POMO | $7.910 \pm 0.055$ | $7.074 \pm 0.010$ | $12.766 \pm 0.358$ | $10.964 \pm 0.171$ |
| | Sym-NCO | $7.035 \pm 0.209$ | $6.334 \pm 0.045$ | $10.776 \pm 0.362$ | $9.159 \pm 0.056$ |
| | SRT (ours) | $\mathbf{6.450 \pm 0.053}$ | $\mathbf{6.038 \pm 0.005}$ | $\mathbf{9.521 \pm 0.098}$ | $\mathbf{8.573 \pm 0.019}$ |
| CVRP | AM Rollout | $13.366 \pm 0.199$ | $11.921 \pm 0.026$ | $23.414 \pm 0.238$ | $19.088 \pm 0.232$ |
| | POMO | $13.799 \pm 0.310$ | $12.661 \pm 0.065$ | $22.939 \pm 0.245$ | $20.785 \pm 0.403$ |
| | Sym-NCO | $13.406 \pm 0.204$ | $12.215 \pm 0.124$ | $21.860 \pm 0.422$ | $18.630 \pm 0.106$ |
| | SRT (ours) | $\mathbf{12.922 \pm 0.071}$ | $\mathbf{11.721 \pm 0.093}$ | $\mathbf{21.582 \pm 0.149}$ | $\mathbf{18.304 \pm 0.109}$ |

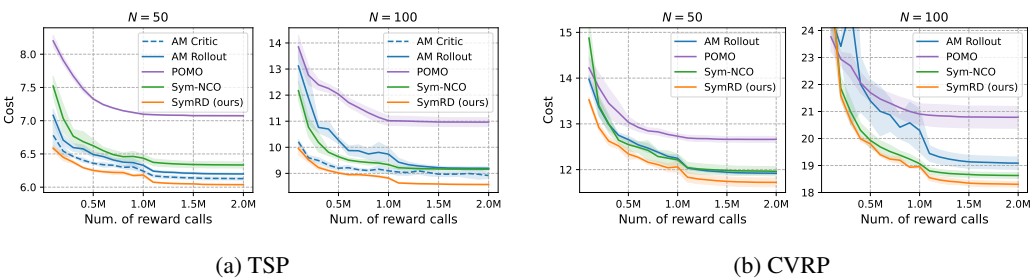

(a) TSP                (b) CVRP

Figure 10: Validation cost over computation budget on euclidean CO problems.

## D.4 EXPERIMENTS ON VARIOUS SYNTHETIC CO PROBLEMS

The experiments in this section cover various sample-efficient tasks in Euclidean and non-Euclidean combinatorial optimization. Note that we assume the expensive black-box reward function in sample-efficient tasks. In Euclidean CO tasks, the features of variables, such as their two-dimensional coordinates, satisfy Euclidean conditions (e.g., cost coefficients are defined as Euclidean distances). On the other hand, non-Euclidean CO problems lack these constraints, necessitating the encoding of higher-dimensional data, such as a distance matrix.

### D.4.1 EUCLIDEAN CO PROBLEMS

**Experimental settings.** We select two representative routing tasks – the travelling salesman problem (TSP) and the capacitated vechile routing problem (CVRP) with 50 and 100 customers. The CVRP assumes multiple salesmen (i.e., vehicles) with limited carrying capacity; thus, if the capacity is exceeded, the vehicle must return to the depot. For base DRL methods, we employ the best-performing DRL methods, AM for TSP and Sym-NCO for CVRP. We follow reported hyperparameters for the model in their original paper.[6]

**Results.** The results in Table 9 and Figure 10 indicate that SRT consistently outperforms baseline methods in terms of achieving the lowest cost over the training budget. Note that ours employs the AM with critic baseline for TSP and Sym-NCO with the reduced number of augmentations for CVRP. As depicted in Table 9, the most significant improvement over the base DRL models is observed in TSP100, with a percentage decrease of 3.86%, and CVRP50, with a percentage decrease of 4.04%. While POMO and Sym-NCO consider the symmetric nature of CO, the required number of samples cancels out the benefits. In contrast, our method utilizes the symmetric pseudo-labels generated via the training policy for free, enabling the policy to explore the symmetric space without increasing the number of required samples. As a result, SRT successfully improves sample efficiency.

---

[6]Sym-NCO: https://github.com/alstn12088/Sym-NCO

Table 10: Experimental results on sample efficient non-Euclidean CO problems.

| | Method | $N = 50$ | | $N = 100$ | |
|---|---|---|---|---|---|
| | | $K = 200K$ | $K = 2M$ | $K = 200K$ | $K = 2M$ |
| ATSP | MatNet-Fixed | $3.139 \pm 0.024$ | $2.000 \pm 0.002$ | $4.400 \pm 0.040$ | $3.227 \pm 0.016$ |
| | MatNet-Sampled | $3.235 \pm 0.021$ | $2.019 \pm 0.005$ | $4.324 \pm 0.036$ | $2.915 \pm 0.040$ |
| | SRT (ours) | $\mathbf{2.845 \pm 0.039}$ | $\mathbf{1.945 \pm 0.003}$ | $\mathbf{3.771 \pm 0.012}$ | $\mathbf{2.513 \pm 0.022}$ |
| FSSP | MatNet-Fixed | $56.350 \pm 0.170$ | $55.341 \pm 0.118$ | $96.461 \pm 0.206$ | $95.107 \pm 0.072$ |
| | MatNet-Sampled | $56.347 \pm 0.234$ | $55.172 \pm 0.032$ | $96.256 \pm 0.140$ | $94.978 \pm 0.055$ |
| | SRT (ours) | $\mathbf{56.104 \pm 0.125}$ | $\mathbf{55.110 \pm 0.061}$ | $\mathbf{96.030 \pm 0.132}$ | $\mathbf{94.934 \pm 0.051}$ |

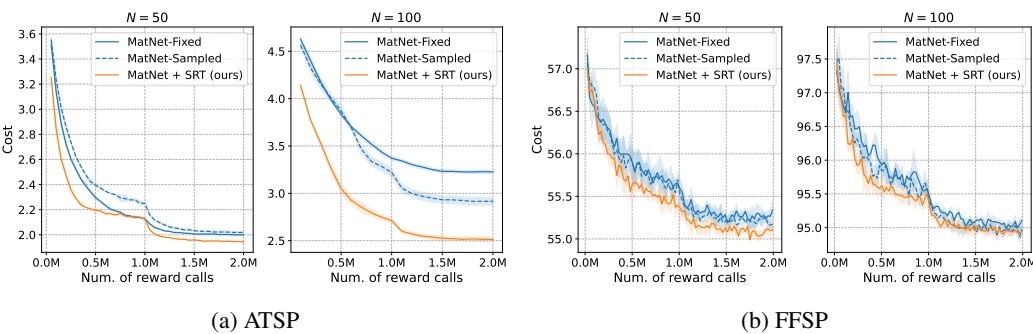

Figure 11: Validation cost over computation budget on non-Euclidean CO problems.

### D.4.2 NON-EUCLIDEAN CO PROBLEMS

**Experimental settings.** Based on the work of Kwon et al. (2021), we have selected two benchmark tasks, namely the asymmetric TSP (ATSP) and flexible flow-shop scheduling problems (FSSP). The ATSP is non-Euclidean TSP where the distance matrix could be non-symmetric, i.e., $\mathrm{dist}(i,j) \neq \mathrm{dist}(j,i)$, where $i$ and $j$ indicate cities. The FSSP is an important scheduling problem that assigns jobs to multiple machines to minimize total completion time. As a baseline, we employ Matrix Encoding Network (MatNet) proposed to solve non-Euclidean CO.[7] We compare ours with two versions of MatNet: MatNet-Fixed and MatNet-Sampled. MatNet-Fixed, the original version, explores $N$ heterogeneous starting points of trajectories, while MatNet-Sampled explores less than $N$ number of multiple trajectories with sampling strategy.

**Results.** The superior performance of SRT over MatNet-Fixed and MatNet-Sampled is demonstrated in both Table 10 and Figure 11. We employ MatNet-Sampled as a base DRL method for both tasks and use the same number of multi-starting in ours and MatNet-Sampled. Notably, SRT outperforms MatNet-Sampled by a significant margin in the case of ATSP, with a performance gap of about 12% at $N = 100, K = 200K$, where SRT achieves 3.771 and MatNet-Sampled achieves 4.324.

---

[7]MatNet: https://github.com/yd-kwon/MatNet

# E    FURTHER RELATED WORKS

## E.1    DEEP REINFORCEMENT LEARNING FOR COMBINATORIAL OPTIMIZATION

Deep reinforcement learning (DRL) has emerged as a promising methodology for solving combinatorial optimization. Especially by selecting actions sequentially, i.e., in a constructive way, DRL policies generate a combinatorial solution. This approach is beneficial to producing feasible solutions that satisfy the complex constraints of CO by restricting action space using a masking scheme (Kool et al., 2018). The foundational work of Bello et al. (2017) introduced the actor-critic method for training PointerNet (Vinyals et al., 2015) to solve TSP and the knapsack problem. Subsequently, several works were proposed to extend PointerNet into a Transformer-based model (Kool et al., 2018; Xin et al., 2021a) especially for routing problems. Building upon the success of the attention model (AM; Kool et al., 2018), Kwon et al. (2020), and Kim et al. (2022b) suggested enhanced reinforcement learning techniques by employing a precise baseline for REINFORCE based on symmetries in CO problems. On the other hand, various works have been suggested to solve broader ranges of CO problems (Khalil et al., 2017; Ahn et al., 2020; Zhang et al., 2020; Jiang et al., 2021; Kwon et al., 2021; Kim et al., 2021; Park et al., 2023; Zhang et al., 2023)

Several studies have proposed to address challenges such as distributional shift and scalability (Hottung et al., 2021; Li et al., 2021; Ma et al., 2021; Choo et al., 2022; Qiu et al., 2022; Son et al., 2023; Ma et al., 2023; Jiang et al., 2023). It is noteworthy that besides the constructive approach, there is another stream, the improvement heuristic style (Hottung & Tierney, 2020; Xin et al., 2021b; Kim et al., 2022a; Ye et al., 2023), though such studies fall outside our research scope. Our research goal is to enhance the sample efficiency of constructive DRL methods for CO; the sample efficiency has been comparatively less explored in contrast to issues such as distributional shift and scalability in DRL for CO literature. This study offers an orthogonal but generally applicable approach to existing works in the field.

## E.2    BLACK-BOX COMBINATORIAL OPTIMIZATION

Building on on the great success of Bayesian optimization (BO) in black-box optimization (Bliek et al., 2023; Irurozki & López-Ibáñez, 2021; Lindauer et al., 2022), several works were suggested to apply BO to combinatorial decision. Combinatorial Bayesian optimization solves a bi-level optimization where the upper problem is surrogate regression, and the lower problem is acquisition optimization. In the context of combinatorial space, the acquisition optimization is modeled as quadratic integer programming problems, which is NP-hard (Baptista & Poloczek, 2018; Deshwal et al., 2022).

To address the NP-hardness of acquisition optimization, various techniques have been proposed, including continuous relaxation Oh et al. (2019), sampling with simulated annealing (Deshwal et al., 2022), genetic algorithms (Moss et al., 2020; Tripp et al., 2021), and random walk explorer (Korovina et al., 2020). While these methods have demonstrated competitive performance in lower-dimensional tasks like neural architecture search (NAS), they often demand substantial computation time, particularly in molecular optimization tasks (Gao et al., 2022).

As mentioned in (Deshwal et al., 2022) and (Gao et al., 2022), one of the alternative approaches is to utilize a variational auto-encoder (VAE; Kingma & Welling, 2014) to map the high-dimensional combinatorial space into "compact" continuous latent space to apply BO, like (Gómez-Bombarelli et al., 2018). Though this approach has shown successful performance in molecular optimization, it also introduces variational error since VAE maximizes a lower bound of the likelihood, known as evidence lower bound (ELBO), not directly maximizes the likelihood.

