# OpenReview forum: "Enhancing Sample Efficiency in Black-box Combinatorial Optimization via Symmetric Replay Training"
_ICLR.cc/2024/Conference — Submitted to ICLR 2024_

### Official Review · Reviewer_vaeD · 2023-10-31

**Soundness:** 3 good
**Presentation:** 4 excellent
**Contribution:** 4 excellent
**Rating:** 8
**Confidence:** 3

**Summary:**

This paper focuses on combinatorial black box optimization (or BBO) problems where candidate solutions need to be generated with a reinforcement learning (RL) policy, and the reward is episodic, and only available once a complete solution is presented to the reward oracle. In this setup, the authors leverage the fact that, in certain problems, distinct action sequences (starting from the same initial state) can lead to the same candidate solution, and make an additional assumption that, given a certain high reward action sequence, such symmetric transformations can be made easily/efficiently. The authors augment the traditional reward-maximizing training of a policy with a ``symmetric replay training'' or SRT, where the policy is additionally trained to imitate symmetric transformations of high rewarding action sequences. This allow the policy training to be more sample efficient by exploring different parts of the action space without needing to query the reward oracle (since the reward is already known).

This method is RL method agnostic, and is added to 5 different RL methods (A2C, PG-Rollout, PPO, GFlowNet, REINVENT), and evaluated across 1 synthetic task, and 2 real world benchmarks. In almost all cases, the proposed SRT significantly improves the sample efficiency of the base methods to different extents (in a couple of cases, SRT is unable to improve upon REINVENT).

**Strengths:**

The paper is very well presented, and it is easy to follow the proposed idea and the subsequent empirical analysis. The authors clearly explain each step and each experiment (with the corresponding tasks and baselines).

The proposed idea of leveraging symmetry in the solution space is a simple but very useful one. This leads to a general method agnostic technique to improve the sample complexity of **any RL-based combinatorial BBO solver**.

The empirical evaluation highlights the wide applicability of the proposed SRT scheme, and the gain SRT provides almost across the board, making it a critical part of any future RL-based solution where such symmetry preserving transformations are readily available.

**Weaknesses:**

Based on the presentation, and the literature review, it is not clear how this method compares to other combinatorial BBO solvers that do not rely on a sequence of actions from a policy to create a candidate solution. Examples of these include various versions of Bayesian Optimization [A, B, C, D].


> - [A] Baptista, Ricardo, and Matthias Poloczek. "Bayesian optimization of combinatorial structures." International Conference on Machine Learning. PMLR, 2018.
> - [B] Oh, Changyong, et al. "Combinatorial bayesian optimization using the graph cartesian product." Advances in Neural Information Processing Systems 32 (2019).
> - [C] Deshwal, Aryan, et al. "Bayesian optimization over permutation spaces." Proceedings of the AAAI Conference on Artificial Intelligence. Vol. 36. No. 6. 2022.
> - [D] Dadkhahi, Hamid, et al. "Combinatorial black-box optimization with expert advice." Proceedings of the 26th ACM SIGKDD International Conference on Knowledge Discovery & Data Mining. 2020.

The need for easily accessible symmetric transformations limit the scope of this BBO solver for BBO problem. On a related note (and not as a weakness), this seems more of an instance of "grey-box" combinatorial optimization instead of black-box because we are leveraging a lot of information regarding the solution space even if we do not have access to an analytic objective function.

- (Minor suggestion) It might be better to put all plots in Figure 3 on the same scale if the reader has to compare them side-by-side

**Questions:**

- How does this RL-based combinatorial BBO solution relate to combinatorial BBO problem handled by Bayesian Optimization schemes such as COMBO [B]? In their case, they would never propose the same solution twice (which is something any RL-based scheme might be producing, and SRT is removing this repetition). And if external information regarding the symmetries between solutions are available, that can be utilized (as is done in this paper) to generate new simulations and/or skip repetitions.
- Are there any existing literature on RL based BBO that discusses or brings up symmetries between action sequences (not necessarily leverages them)?
- Isn't TSP a problem where we can get a reward after every action (and the final reward is the sum of the partial rewards)? What happens if we compare the proposed scheme to such a setup with a RL-based solution?
- Are we looking at symmetric action sequences leading to the same solution $\mathbf{x}$, or are we talking about action sequences that lead to different solutions $\mathbf{x}'$ but $f(\mathbf{x}) = f(\mathbf{x}')$? The TSP example given on page 3 after Definition 1 seem to fit this latter setup. If it is infact the latter, how does that effect the proposed scheme and the related motivations?
- How is the shifting related to the starting point? Will a different sequence of actions from the same starting point lead to a solution with the same cycle (and hence value) but different order? I agree that the reversed order does lead to the same solution with the same initial state, but I am not sure about the other scenarios.
- I am not sure where the 20+% improvement is coming in for PPO+SRT over PPO with K=100k (as discussed in Section 4.1). Table 1 lists PPO at $6.77 \pm 0.12$ vs $6.71 \pm 0.02$ for PPO+SRT, which is more like a 1% improvement. Am I looking at the wrong results?
- In the presence of the symmetry, and the access to symmetric transformations, it seems that we should always see improved sample efficiency (or at least no degradation). Under what conditions might we expect degradation (as we see with 2 cases in Table 3 with REINVENT)?

---

> ### Author Response · Authors · 2023-11-17
> **Responses for the reviewer vaeD**
>
> Thanks for the valuable comments.
>
>
> ### W1: not clear how this method compares to other combinatorial BBO
>
> **Answer**
> *First of all, our research **focuses on constructive RL methods for combinatorial optimization** where the function evaluation is expensive. Based on the suggested paper, we clarify our research scope and why we focus on constructive RL methods.*
>
> Combinatorial Bayesian optimization solves a bi-level optimization where the upper problem is surrogate regression, and the lower problem is acquisition optimization. In the context of combinatorial space, the acquisition optimization is modeled as quadratic integer programming problems, which is NP-hard [A, C]. Note that the surrogate regression has cubic complexity when using the Gaussian process; thus, we need to solve problems with cubic complexity and NP-hardness every iteration.
>
> To address the NP-hardness of acquisition optimization, various techniques have been proposed, including continuous relaxation [B], sampling with simulated annealing [C], genetic algorithms (BOSS [1] and GPBO [2]), and random walk explorer (ChemBO [3]). While these methods have demonstrated competitive performance in lower-dimensional tasks like neural architecture search (NAS), they often demand substantial computation time, particularly in molecular optimization tasks [4]. In Appendix D.3, we have included the results of GPBO for comparison with our approach. Note that GPBO requires the design of domain-specific heuristic algorithms, which limits the extendability to other tasks.
>
> As pointed out in [C], one of the alternative approaches is to utilize a variational auto-encoder (VAE) to map the high-dimensional combinatorial space into “compact” continuous latent space to apply BO, like [5]. Though this approach has shown successful performance in molecular optimization, it also introduces variational error since VAE maximizes a lower bound of the likelihood, known as evidence lower bound (ELBO), not directly maximizes the likelihood.
>
> *In addition, based on the suggested research, we added the literature review about black-box combinatorial optimization in Appendix E and the experimental results of Bayesian optimization on the molecular optimization tasks. The results are in Appendix D.3*.
>
> ### W2: grey-box problems rather than black-box
>
> **Answer**
> We agree with your comment that our target problems are grey-box rather than black-box. Our target problem is the CO problem, whose objective function evaluation is expensive. One example can be a CO problem with a black-box objective function and white-box constraints. The title and manuscript have been revised, as mentioned in the general response.
>
>
> ### Minor suggestion:
> We adjusted the y-scale of plots in Figure 3 and also in Figure 6. Thanks for the suggestion.
>
> ### Q1: Is it possible to utilize symmetries in other methods for BBO?
>
> **Answer**: In reinforcement learning, a policy constructs a new solution by sequentially selecting actions from an empty solution. Consider the case of determining a set with three components, where a solution is constructed as follows: [ ] -> [a] -> [a, b] -> [a, b, c]. This sequential construction gives rise to multiple trajectories (e.g., [] -> [b] -> [b, c] -> [b, c, a]), and our approach leverages this symmetry.
> On the other hand, in Bayesian optimization, new solutions are suggested by exploring the solution space starting from an initial solution. For instance, by employing Gibbs sampling within a 1-Hamming ball on discrete space, e.g., [a, b, c] -> [d, b, c] -> [d, b, a], there are no such symmetries that we utilize in RL.
>
>
> ### Q2: Symmetries in RL-based BBO
>
> **Answer:** In DevFormer, they introduce an additional loss term to consider solution symmetries; however, these symmetries are not induced in a structured way. On the other hand, GFlowNet introduces a new kind of loss function that considers solution symmetries by modeling the sequential decision-making in CO on a directed acyclic graph (DAG) and considering flows on DAG as a policy. Our two-step learning procedure can be used along with these DRL methods, and SRT can further induce solution symmetries effectively.
> It is noteworthy that our algorithm can also easily cooperate with symmetric architectures like SE3 Transformer [6] and EGNN [7], which exhibit architectural symmetries in state-level representation (e.g., rotation of an image yields the same representation). Since these models do not consider solution symmetries related to the action trajectories, further learning processes like SRT are required to induce solution symmetries.

---

> > ### Author Response · Authors · 2023-11-21
> > **Gentle reminder for further discussion**
> >
> > Dear Reviewer vaeD,
> >
> > Thank you for sharing your thoughtful comments. We kindly encourage you to review our response to ensure that we have effectively addressed the concerns and questions raised in your comments. We are open to further discussion and would appreciate your feedback. If you have any additional questions or unresolved issues, please feel free to let us know.

---

> > > ### Comment · Reviewer_vaeD · 2023-11-21
> > > **Thank you for the detailed response**
> > >
> > > The authors have addressed my questions and concerns, and consequently, I have raised my score.
> > >
> > > A few further comments and thoughts:
> > >
> > > - I appreciate the detailed positioning against Bayesian Optimization based black-box Combinatorial Optimization. The new results in Section D.3, Table 8 are very helpful. If I understand it correctly (given the response to Q7), the molecular optimization benchmarks is the most challenging setup in terms of exploiting symmetries (possibly because of the restriction on the API, but maybe other reasons as well).  In this case, the symmetry exploiting REINVENT+SRT (slightly) outperforms the Bayesian Optimization based scheme. While the authors claim that the development of efficient acquisition optimization requires domain specific knowledge, and is critical to the final performance, it is not clear why that is not **also the case with SRT**. Obtaining symmetric transformations would also require a lot of domain specific knowledge, and would have to change as the problem domain changes. SRT itself is not a domain agnostic scheme -- we need domain knowledge (which is why I referred to it as a grey box problem).
> > > - Thank you for the detailed description of the TSP setup and the symmetries therein. That is very helpful. Just as a note, this example and discussion also highlights how the symmetric transformations are domain specific.
> > > - Regarding Q7, where the authors discuss possible reasons for the underperformance of SRT. If overfitting is an issue when there are not sufficient symmetries to exploit, then it might be useful to leverage SRT in a more adaptive way (rather than always). The authors bring up the "frequency of conducting SRT". How might one go about it in a problem dependent manner to avoid underperforming the base model?
> > > - Finally, it appears that some of the reported numbers have changed (without a blue mark). For example, the numbers for A2C in Table 1 has changed. I did not check all the numbers though so I do not know all the places the change happened.

---

> > > > ### Author Response · Authors · 2023-11-22
> > > >
> > > > Dear reviewer vaeD,
> > > >
> > > > We appreciate the positive feedback and further comments.
> > > >
> > > > We agree that SRT also requires domain knowledge. It is noteworthy that symmetries employed in SRT are comparatively simple to identify. Leveraging the observation that a constructive RL policy produces a sequence (permutation) despite the solution being a set (combination), general rules can apply. For instance, permuting action orders gives symmetric trajectories in the decap placement problem. This rule can be applied to the knapsack problems, aiming to select items under a fixed budget to maximize profit. In another case, permuting indices can be used. By permuting vehicle and machine indices, symmetric trajectories are found in the capacitated vehicle routing problem and the flexible flow-shop scheduling problem. Once the symmetries are found, training a DRL policy with SRT is easy; we believe this is simpler than designing chromosomes and cross-over operations in a genetic algorithm.
> > > >
> > > > Secondly, to leverage SRT in a more adaptive way, we suggested adjusting the frequency of conducting SRT. We have observed that higher performance is achieved when an SRT update is conducted every five RL loops in some tasks in molecular optimization. In this case, we set the frequency in a trial-and-error manner, but there could be a more effective way. We believe this is one of the good further works.
> > > >
> > > > Lastly, while conducting additional experiments, we found that the results of A2C (+ SRT) had been reported based on the earlier results. (All experiments had been reconducted when the code was merged; each task has different dependencies, so the environment differed from the earlier experiments.) That is the only change in the numbers. We apologize for missing the blue color on it.
> > > >
> > > > Again, thanks for your helpful feedback.

---

> ### Author Response · Authors · 2023-11-17
> **Responses for the reviewer vaeD (2)**
>
> ### Q3: What happens if we use intermediate reward to train RL policy in TSP?
>
> **Answer:** First of all, as a TSP solution, we need to find a Hamiltonian cycle covering all cities. The cost of the solution can vary based on the last segment, which involves returning to the starting point. Note that the last segment is automatically defined on the previous sequence. In other words, the reward is episodic. Training the model with intermediate rewards could introduce bias into the return value, so performance is insufficient. Even in heuristic algorithms for TSP, the farthest insertion, which starts to form a new cycle by inserting the farthest city, outperforms other heuristics. Therefore, the RL policy should be trained using the Monte Carlo method with episodic rewards to ensure unbiased learning.
>
>
> ### Q4, Q5: about symmetric trajectories in TSP
>
> **Answer:** In the TSP, the decision variable $x_{ij}$ takes the value of 1 if city $j$ is visited immediately after city $i$. For instance, when considering four cities, the sequences 1-2-3-4-1 and 2-3-4-1-2 represent the same solution (i.e., x is the same). This implies an identical cycle and, apparently, the same decision variables in both sequences. This is what we intended in the example on page 3. Note that a TSP solution (there is no specific starting point) is a Hamiltonian cycle. We have made revisions to the example in the manuscript.
>
>
> ### Q6: 20+% improvement is coming in for PPO+SRT over PPO **
>
> **Answer:** There are some typos; we revised the sentence as follows: The most substantial improvement facilitated by SRT is observed in the case of GFlowNet, where a cost reduction of 3.76% is achieved when the sample size K=100K.
>
> ### Q7: When does the performance degradation happen?
>
> **Answer:** Based on the experimental results in Figure 5 (the replay loops are increased), the model can suffer from overfitting when there are not enough symmetries to utilize. For example, unlike TSP, we cannot explicitly get symmetric solutions in the PMO benchmark. The symmetric transformation highly depends on the API; thus, it is hard to explore symmetric space effectively (the maximum entropy is achieved with the uniform symmetric transformation policy according to Theorem 1), resulting in performance degradation due to the early convergence in some cases. This can be mitigated by designing a more precise symmetric transformation policy or adjusting the frequency of conducting SRT.
>
> ---
> ### Reference
>
> [1] Moss et al. “BOSS: Bayesian optimization over string spaces.” NeurIPS, 2020.
>
> [2] Austin Tripp, Gregor NC Simm, and José Miguel Hernández-Lobato. “A fresh look at de novo molecular design benchmarks.” NeurIPS 2021 AI for Science Workshop, 2021.
>
> [3] Korovina et a. “ChemBO: Bayesian optimization of small organic molecules with synthesizable recommendations.” AISTATS, 2020.
>
> [4] Gao et al. "Sample efficiency matters: a benchmark for practical molecular optimization." NeurIPS, 2022.
>
> [5] Gomez-Bombarelli et al. “Automatic Chemical Design using a Data-driven Continuous Representation of Molecules.” ACS central science, 2018.
>
> [6] Fuchs, Fabian, et al. "SE(3)-Transformers: 3D roto-translation equivariant attention networks." NeurIPS, 2020.
>
> [7] Satorras, Vıctor Garcia, Emiel Hoogeboom, and Max Welling. "E (n) equivariant graph neural networks." International conference on machine learning. PMLR, 2021.

---

### Official Review · Reviewer_j1D8 · 2023-11-01

**Soundness:** 2 fair
**Presentation:** 2 fair
**Contribution:** 2 fair
**Rating:** 5
**Confidence:** 4

**Summary:**

This paper targets expensive combinatorial black-box optimization problems (with limited function evaluations). To enhance the sample efficiency of deep reinforcement learning (DRL), it introduces the Symmetric Replay Training (SRT) method, which suggests training the DRL agent by alternating between the conventional RL loss and a symmetric loss. The symmetric loss is designed to mimic previously generated trajectories that yield high rewards, while taking advantage of the solution-symmetric characteristics inherent in the combinatorial space. Results verified that SRT can boost the sample efficiency of the base DRL method.

**Strengths:**

From a technical perspective, the proposed SRT method appears to be valid. The emphasis on improving sample efficiency is important. Moreover, the method has been tested across three distinct categories of optimization problems, showcasing its versatility and effectiveness.

**Weaknesses:**

The paper seems to have a scattered focus. While the abstract suggests a focus on expensive black-box combinatorial optimization, the studied TSP problem and most leveraged baselines (like PPO, AM, GFlowNets) neither belongs to black-box optimization nor expensive optimization problems/algorithms. The paper also lacks a comparison with other methods specifically designed for expensive black-box optimization tasks, such as [1-4]. This makes positioning the paper within the literature quite challenging.

Meanwhile, reinforcement learning methods, especially on-policy ones like PPO, A2C, are criticized for their general sample efficiency. It remains unclear why on-policy RL is necessary for solving expensive black-box optimization problems.  This paper also neglects the comparison with other recent research that explicitly focused on improving the sample efficiency of DRL based on data augmentation and experience replay.

To me, it seems slightly unclear which community the paper could contribute: Is it DRL for CO, DRL's sample efficiency, expensive optimization, or expensive black-box optimization?

In addition, I have concerns about whether all expensive black-box optimizations can identify appropriate symmetries.
***
[1] Benchmarking Surrogate-based Optimisation Algorithms on Expensive Black-box Functions

[2] Unbalanced mallows models for optimizing expensive black-box permutation problems

[3] Evolutionary Computation for Expensive Optimization: A Survey

[4] SMAC3: A Versatile Bayesian Optimization Package for Hyperparameter Optimization

**Questions:**

1. Considering the symmetric loss, should we factor in importance sampling because methods like PPO and A2C are on-policy RL?
2. The paper would benefit from pseudo-code to clearly describe the algorithm. Presently, it's unclear how SRT integrates with PPO, A2C and other baselines.
3. How optimal are the results in Table 1 for TSP? How good are the SRT to advance the state-of-the-art of the expensive black-box combinatorial optimization?
4. Why do the results in Table 3 could differ from those in "Sample efficiency matters: a benchmark for practical molecular optimization"?
5. In Figure 3, PPO demonstrates a significant variance. This raises concerns about whether the hyper-parameters or other settings for PPO have been correctly set.

---

> ### Author Response · Authors · 2023-11-17
> **Responses for the reviewer j1D8 (1)**
>
> Thanks for the insightful comments.
>
> ***Summary:** Our research goal is to enhance the sample efficiency of **deep RL for Combinatorial Optimization**, which is crucial since practical CO problems often involve computationally expensive objective functions, e.g., black-box function. While various methods have been suggested to address black-box optimization effectively, applying them directly to combinatorial optimization poses difficulties (details will be discussed later). On the other hand, RL-based methods also have shown promising performance in CO. However, they leverage numerous samples, which is unaffordable when the function evaluation is expensive. Specifically, **our research focuses on how to discover and induce effective inductive bias based on the nature of CO**. In the following responses, we further discuss why we mainly use on-policy RL in CO and address each concern in detail.*
>
> ----
>
> ### W1: The paper seems to have a **scattered focus**
>
> ### W5: It seems slightly **unclear which community the paper could contribute**.
>
> **Answer**
> We agree that there were some misalignments in the manuscript. We believe our work contributes to **deep RL for Combinatorial Optimization** fields that require sample efficiency. We have revised the title and abstract for better alignment in the manuscript.
>
>
> ### W2: the baselines are not for black-box CO and there is no comparison with methods for black-box optimization.
>
> **Answer**
>
> Thanks for the insightful comments and suggestions. We have looked through the literature you recommended and added the literature review about black-box combinatorial optimization based on the suggested research in Appendix E.
>
> First of all, we additionally provide the results of a BO-based (GPBO) and a genetic algorithm (Graph GA) according to your suggestion ([1,2,4] - Bayesian optimization, [3] - evolutionary computation). The table shows the average AUC with three independent seeds. We provide the detailed results in Appendix D.3.
>
> |  | GPBO | Graph GA | REINVENT | REINVENT + SRT |
> | --- | --- | --- | --- | --- |
> | Avg. AUC | 0.609 | 0.619 | 0.613 | **0.633** |
>
> Here, we would like to discuss why we solve CO problems using reinforcement learning, especially when the function evaluation is expensive.
>
> -----
>
> **Challenges of applying evolutionary computation algorithms to CO.** The paper [3] introduces evolutionary computation (EC) algorithms, including genetic algorithms (GA). While EC algorithms have a well-established reputation for delivering powerful performance across diverse tasks, they do exhibit certain limitations. As mentioned in [3], *designing novel operators, which greatly affects EC algorithms' performance, requires domain knowledge.* This implies that careful algorithm design is necessitated whenever there is a change in tasks. For example, though Graph GA achieves promising performance, performance in other CO problems will vary depending on how well-designed crossover and mutation operators are.
>
> **Challenges of Bayesian Optimization Algorithms to CO.** As indicated in [1,2,4], Bayesian optimization (BO) is a promising method for solving black-box optimization problems. The key idea is to employ a surrogate model, usually a Gaussian Process, and optimize an acquisition function built upon the surrogate model to draw samples. BO has demonstrated its powerful performance in row-budget regimes, especially in continuous space.
> However, in the case of combinatorial spaces, constructing an accurate surrogate model poses a substantial challenge due to its non-smoothness. Furthermore, the acquisition optimization is modeled as quadratic integer programming problems, which is NP-hard [5]. Strategies to address this NP-hardness involve the utilization of variational auto-encoders [6] or continuous relaxation [7]. Yet, these approximations can result in significant performance degradation.
>
> **Sampling in high-dimension is hard.** It is challenging to sample solutions in CO, as they are high-dimensional. To effectively sample a solution, the joint distribution can be factorized as follows.
> $$x\sim p(x) = \prod_{t=1}^T p(x_i|s_t)$$
>
> Sampling a solution from the factorized distribution can also be regarded as sequential decision-making, which RL effectively models.
>
> **Benefits of solving CO with RL.** In RL for CO, solutions are constructed by sequentially deciding on $x_i$. This constructive policy is advantageous in satisfying the constraints of CO by restricting decisions that lead to infeasible solutions at each step. The fulfillment of constraints is a key property in CO.
>
> Another benefit of this approach is that we can construct a general solver that addresses various target problem instances instead of exclusively seeking optimal solutions for individual target instances, i.e.,
> $$x \sim p(x|c) = \prod_{t=1}^T p(x_i|s_t, c)$$
>
> In this research, we train a general solver for the traveling salesman problems and decap placement problems in Sections 4.1 and 4.2.

---

> ### Author Response · Authors · 2023-11-17
> **Responses for the reviewer j1D8**
>
> ### W3: It remains unclear why on-policy RL is necessary for solving expensive black-box optimization problems.
>
> **Answer**
> Due to the combinatorial nature, 1) rewards are episodic, and 2) value function estimation is notoriously hard. Here, episodic reward means we cannot evaluate objective function (i.e., reward) in the middle of constructing solutions. Therefore, Monte Carlo-based methods, like REINFORCE, have been employed in DRL for CO literature. **When we use Monte Carlo, importance sampling is required for off-policy methods, leading to high variance.** As evidence, the empirical results show that REINVENT, an on-policy method, outperforms off-policy methods such as GFlowNets and DQN in the PMO benchmark. Though on-policy is known to give superior performance in CO, **we also verified our method with an off-policy method, GFlowNet.** We added supplement explanations for GFlowNet in the manuscript.
>
> ### W4: comparison with data augmentation and experience replay
>
> **Answer**
>
> **(Augmentation)** We agree that data augmentation like cropping [8, 9], noise injection [8, 10], and Mixup [10, 11] can improve sample efficiency. However, these methods focus on visual RL scenarios (continuous space), hence not applicable to CO. In the case of TSP in Euclidean space, POMO and Sym-NCO propose augmenting graph node features by rotating coordination values. As illustrated in Appendix D.4, SRT improves the sample efficiency of Sym-NCO. Note that augmentations in Sym-NCO are done in input space, while ours utilize symmetries in output (action trajectories) space.
>
> **(Experience Replay)** We have conducted additional experiments with A2C on TSP to compare ours to experience replay (ER) as follows. We set the buffer size to 10000, which is the same as the epoch size. Note that when we use experience replay to on-policy, importance sampling is required as follows.
>
> $$E_{a \sim p(\cdot|s_1)}[R(x)] = E_{a \sim q(\cdot|s_1)}[ \frac{p(a|s_1)}{q(a|s_1)}R(x) ] = E_{a \sim q(\cdot|s_1)} [\prod_{t=1}^{T}\frac{p(a_t|s_t)}{q(a_t|s_t)}R(x)],$$
> where $p(\vec{a})$ is the training policy and $q(\vec{a})$ is the behavior policy that made the experiences in the replay buffer, and $x=C(\vec{a})$. The importance weight terms of $\prod_{t=1}^{T}\frac{p(a_t|s_t)}{q(a_t|s_t)}$ gives high variances to estimate objective so that gives poor performances on replay training as shown in the following table. The detailed experimental settings and results are also provided in Appendix D.2.
>
> | Method | 100K | 2M |
> | -------- | -------- | -------- |
> | A2C      | 6.630 +- 0.037 | 6.115 +- 0.009 |
> | A2C + ER (size = 20) | 7.085 +- 0.326 | 6.156 +- 0.033 |
> | A2C + ER (size = 100) | 7.073 +- 0.293 | 6.139 +- 0.010 |
> | A2C + SRT (ours) | *6.560 +- 0.051* | *6.037 +- 0.005* |
>
>
> The results show that experience replay can degrade the DRL method due to the high variance introduced by the importance sampling. In contrast, our method does not suffer from the importance sampling because SRT updates the policy to maximize the likelihood of symmetric samples.
>
> Additionally, we provide the results on GFlowNet, off-policy RL, on molecular optimization tasks. Here, the experience replay improves the sample efficiency of GFlowNet, but our method shows more effective results. Average AUC with three independent seeds is reported.
>
> | GFlowNet | + Experience Replay | + Reward-prioritized ER | + SRT |
> | -------- | -------- | -------- |-------- |
> | 0.436 | 0.491 | 0.497 | *0.511* |
>
> *Note that on-policy RL methods with Monte Carlo estimation have shown notable performance in the CO domain. Though augmentation and experience replay are well-known general approaches to improve sample efficiency, directly applying these into DRL for CO can give insufficient performance gain and even lead to performance degradation.*
>
> ### W6: concerns whether all expensive black-box optimizations can identify appropriate symmetries.
>
> **In general, finding appropriate symmetries can be tricky in back-box optimization. However, when a constructive policy is employed for CO, there are always symmetries we can utilize.**
> From a reinforcement learning perspective, a policy sequentially selects actions to generate a solution. In combinatorial optimization, solutions should be represented in sets, while RL policy generates a sequence (or a graph). Due to the relationship between sets (combination) and sequences (permutation), there must be symmetries. For instance, there are 4! ways to create a set like [A, B, C, D] in sequence. By utilizing this relationship, we can easily find symmetric trajectories whether the objective function is black-box or not.
> Note that the generation process of RL can be canonicalized in a particular order (e.g., from left to right) but introduces a significant bias. Hence, when tackling combinatorial optimization problems, it is crucial to consider symmetries in an appropriate way.

---

> ### Author Response · Authors · 2023-11-17
> **Response for the review j1D8 (3)**
>
> ### Q1: Considering the symmetric loss, should we factor in importance sampling because methods like PPO and A2C are on-policy RL?
>
> We do not need to consider importance sampling because we use imitation learning as replay training, which is an off-policy method. We believe this is the significance of our approach over typical replay training methods for on-policy RL.
>
>
> ### Q2: pseudo-code
>
> **Answer:** We have added a pseudo-code in Appendix B.1.
>
>
> ### Q3: How optimal are the results in Table 1 for TSP?
>
> **Answer:** In the synthetic setting, we assume that the number of reward calls is limited (but, in fact, it can be computed). Therefore, we can get the optimal solutions with off-the-shelf solvers like Concorde. The average optimal cost is 5.696.
>
> ### Q4: Why do the results in Table 3 differ from those in the original paper?
>
> **Answer:** We may use a different set of seeds with the original paper; we use [0, 1, 2, 3, 4] regardless of oracles. The PMO did not report which seeds were used for the main results. We use a different version of PyTDC (Therapeutics Data Commons [12]), which directly affects Oracle score. We use the updated versions (v0.4.0) since the previous version (v0.3.6) gives invalid scores for some tasks.
>
>
> ### Q5: Why do the results of PPO show a high variance?
>
> **Answer:** As provided in Appendix B.2, we have searched (2x3x3) hyper-parameter combinations for tuning PPO. Note that PPO is not a popular choice in the field of neural combinatorial optimization because PPO requires critic estimation for high performance, and estimating the value function for combinatorial optimization is challenging. Additionally, the variance in Table 1 is not that large compared to others, e.g., PG-Rollout with 100K. Most importantly, we use the same hyper-parameters for PPO + SRT.
>
> ----
> ### Reference
> [5] Baptista, Ricardo, and Matthias Poloczek. "Bayesian optimization of combinatorial structures." ICML, 2018.
>
> [6] Gomez-Bombarelli et al. “Automatic Chemical Design using a Data-driven Continuous Representation of Molecules.” ACS central science, 2018.
>
> [7] Oh, Changyong, et al. "Combinatorial bayesian optimization using the graph cartesian product." NeurIPS, 2019.
>
> [8] Laskin et al. "Reinforcement learning with augmented data." NeurIPS, 2020.
>
> [9] Denis Yarats, Ilya Kostrikov, and Rob Fergus. "Image augmentation is all you need: Regularizing deep reinforcement learning from pixels." ICLR, 2020.
>
> [10] Fan et al. "SECANT: Self-expert cloning for zero-shot generalization of visual policies." ICLR 2021.
>
> [11] Zhang et al. "Mixup: Beyond empirical risk minimization." arXiv preprint arXiv:1710.09412, 2017.
>
> [12] Huang et al., "Therapeutics data commons: Machine learning datasets and tasks for therapeutics." NeurIPS Track Datasets and Benchmarks, 2021. (https://tdcommons.ai)

---

> ### Author Response · Authors · 2023-11-21
> **Gentle reminder for further discussion**
>
> Dear Reviewer j1D8,
>
> Thank you for sharing your thoughtful comments. We kindly encourage you to review our response to ensure that we have effectively addressed the concerns and questions raised in your comments. We are open to further discussion and would appreciate your feedback. If you have any additional questions or unresolved issues, please feel free to let us know.

---

> > ### Comment · Reviewer_j1D8 · 2023-11-22
> >
> > I appreciate the authors for their detailed and thorough response, which has addressed some of my concerns. I also noted the updates made to the title and abstract of the paper, shifting its focus towards "Deep Reinforcement Learning for Combinatorial Optimization". This focus necessitates a more comprehensive discussion of related work in the field. Meanwhile, I still believe the paper's technical contribution is somewhat limited and needs justification, especially in terms of its clear contribution to the above filed. Despite these issues, the authors have conducted extensive experiments and demonstrated good performance of their algorithm, which has led me to increase my score to 5. Nonetheless, I remain relatively hesitant to support this paper due to concerns about its potentially limited future impact.

---

> > > ### Author Response · Authors · 2023-11-22
> > >
> > > Dear reviewer j1D8,
> > >
> > > Thank you for the response and positive feedback. To address the concern raised in your additional comments, further related works on deep reinforcement learning (DRL) for combinatorial optimization have been added in Appendix E; please check our revised manuscript. In addition, we would like to discuss how our work can contribute to this field - DRL for CO, in the future as follows.
> > >
> > > Potential Future Impact of This Work:
> > >
> > > 1. **Expandability for future studies.** One of the key strengths of this work is its expendability. The proposed method, replay training with imitation learning using symmetric action trajectories, applies to both on-policy and off-policy reinforcement learning. Furthermore, ours exhibits additional improvements in DRL methods, such as Sym-NCO, considering symmetries. Our work has an impact as our method can easily be applied to various DRL methods for combinatorial optimization in the future.
> > >
> > > 2. **Meaningful take-home message for future research.** In combinatorial optimization, sample efficiency has been less studied than other challenges, such as distributional shift or scalability.  This work brings up the importance of sample efficiency. The proposed method shows that properly exploiting solution symmetries, regarded as a reason for inevitably extended search space, can improve sample efficiency. We believe our study can inspire future work to consider sample efficiency in DRL for CO for better practicality and encourage properly utilizing symmetries in CO.
> > >
> > > Please consider our impact on future studies in this field because this is the first work considering sample efficiency in DRL-based combinatorial optimization. We believe our works can inspire future researchers.

---

### Official Review · Reviewer_t4AA · 2023-11-02

**Soundness:** 4 excellent
**Presentation:** 4 excellent
**Contribution:** 2 fair
**Rating:** 5
**Confidence:** 4

**Summary:**

This paper proposes a method called _symmetric replay training_ (SRT) to improve the sample-efficiency of reinforcement learning algorithms on problems where an agent constructs a final state step-by-step (e.g. building a molecule one atom at a time or placing hardware components on a chip one at at time). Such problems contain a natural combinatorial symmetry where the same final state can be reached by performing the same actions but in a different order. The method proposed in the paper is to supplement normal RL training with maximum likelihood training on transformed versions of the action sequences found during the previous round of RL. Experimentally, the authors show that this helps improve sample-efficiency in 3 problem domains.

**Strengths:**

Overall I thought this paper was well-done.

Strengths:

- The idea is sensible and simple.
- Experimentally, it does seem to provide improvements over the same RL methods done without SRT.
- The paper is well-written and well-presented. This does not feel like a rushed/last-minute submission. I liked the figures and diagrams.
- The experiments are thorough and

**Weaknesses:**

Weaknesses:

- The key idea is, at least in my opinion, kind of obvious. I'm not an RL expert so I can't comment on how previous works in RL have exploited symmetries, but at least in chemistry the existence of these symmetries is well-known and has been exploited in prior work (e.g. the paper "All SMILES Variational Autoencoder for Molecular Property Prediction and Optimization"). To the authors' credit though, I am not aware of work which has done this specifically for RL methods.
- At least for molecules, the experimental results are not as impressive as they may seem. A recent paper showed that tuning genetic algorithms actually can achieve an average score of `0.639` on the PMO benchmark, which is a bit better than the results shown in this paper (http://arxiv.org/abs/2310.09267). Furthermore, the bolding in Table 3 is a bit misleading: they bold the tasks which achieve the best results just on the methods considered, but not necessarily overall SOTA results on the PMO benchmark. For example, drd2, which the authors bolded with a score of `0.960` is beaten by SynNet with a score of `0.969` (Gao et al 2022).

More broadly, in my opinion, this paper's contribution only makes sense because RL methods are being applied to non-sequential problems by introducing some sort of artificial action sequence. To me, this suggests that RL is not the right tool for these kinds of problems. However, this is just my opinion and I did not take this into account when deciding my score.

Overall, I think this paper proposes a sensible method which does improve performance, as shown in a well-designed experimental evaluation. However, the idea is somewhat obvious and only applies to a class of methods which I think are not really well-suited to the problem. Putting these factors together, I feel that a score of 6 is appropriate.

**Questions:**

- In TSP you assume distances are Euclidean. However, is this not an easier subtype of TSP problems? I thought the NP-hard version is occurs when there is no such distance heuristic. I guess it is a toy problem so it doesn't matter much, but it might be more impactful to show result on a harder type of TSP.
- Theorem 1 was unclear to me: what specifically is $p(x|s_1)$? More generally, the importance of this theorem's result was not clear to me...
- Typo: Page 4: "sale of loss function" -> "scale [....]"

---

> ### Author Response · Authors · 2023-11-17
> **Responses for the reviewer t4AA**
>
> Thanks for the valuable comments.
>
>
> **Answer for W1: symmetries in chemical literature**
>
> We agree that there are prior works about symmetries both in RL communities and chemical discovery communities. The major novelty of our works to compare with prior works is we reuse existing samples (which is expensive) given from the RL training phase and exploit prior knowledge of symmetries so that we make "free" samples to make replay training with imitation learning for support RL training more sample efficient. In particular, the paper "All SMILES Variational Autoencoder for Molecular Property Prediction and Optimization" focuses on how to make VAE representations invariant to symmetric transformation while we focus on the architecture agonistic method to enhance sample efficiency of RL training.
>
> **Answer for W2: Experimental results for molecules may not be as impressive compared to the recent research**
>
> We agree about that. On the scientific discovery side, there can be better candidate algorithms, such as a genetic algorithm.
> On the other hand, RL is a general tool that can cope with general-purpose combinatorial discovery and optimization without the need for specific domain knowledge. In addition, we revised the caption of Table 3 to avoid misleading.
>
>
> **Answer for why RL is used**
>
> It is challenging to sample solutions in CO, as they are high-dimensional. To effectively sample a solution, the joint distribution can be factorized as follows.
> $$x\sim p(x) = \prod_{t=1}^T p(x_i|s_t)$$
>
> Sampling a solution from the factorized distribution can also be regarded as sequential decision-making, which RL effectively models. Our research focuses on the auto-regressive RL, which constructs solutions sequentially, as it is advantageous to avoid infeasible solutions. Since CO contains (hard) constraints, generating feasible solutions is important.
> We employed RL-based constructive policies to satisfy feasibility while maintaining computational tractability in high-dimension space. It leads to inevitable order bias and, thus, solution symmetries. Instead of suppressing symmetries, we use them to systematically explore the action trajectory space by replaying free symmetric trajectories. As we do not restrict architecture to induce symmetries, our method flexibly works with various DRL methods.
> In summary, constructive RL policies are advantageous to solve the CO problem effectively. In this study, we propose a generic method that leverages symmetric trajectories induced by introducing an artificial order to enhance sample efficiency in a positive manner.
>
>
> **Questions**
>
> - **Q1: hardness of TSP.** Euclidean TSP is one of the cases of TSP where the distance is defined as Euclidean distance. Generally, the cost function of TSP is defined as $\sum_{i,j} d_{ij}x_{ij}$, and regardless of how to define $d_{ij}$, TSP is NP-hard.
> - **Q2: Unclear explanation of Theorem 1.** $p(x|s)$ is the distribution over the solution. Since we have multiple action-trajectories that give the same solution, $p(x|s_1) = \sum_{a \in A_x} \pi_\theta(a|s_1)$ holds
> Here, $A_x$ denotes the space of trajectory associated with the solution x, i.e., $C(a) = x$ for all $a \in A_x$. This Theorem gives a theoretical background for employing a randomized symmetric transformation policy, which is advantageous. For example, in TSP, there are $2N$ symmetric trajectories. If we use the fixed symmetric transformation, e.g., flip only, the entropy of the policy is bounded. Therefore, to maximize the entropy of the symmetric transformation policy, we uniformly sample the symmetric trajectories over all possible $2N$ trajectories. The experimental results also support this as follows. Also, we revised the manuscript for better readability.
>
> | Method | K=100K | K=1M | K=2M |
> | --- | --- | --- | --- |
> | A2C | 6.630 +- 0.037 | 6.514 +- 0.049 | 6.115 +- 0.009 |
> | + SRT (fixed) | 6.775 +- 0.079 | 6.198 +- 0.024 | 6.069 +- 0.014 |
> | + SRT (ours) | 6.560 +- 0.051 | 6.171 +- 0.025 | 6.037 +- 0.005 |
>
> - **Q3:** We revised the typo.

---

> > ### Comment · Reviewer_t4AA · 2023-11-20
> > **Thanks for the response**
> >
> > I read your response and it answered my questions, although it hasn't really changed my opinion of the paper. For the time being I intend to keep my score.

---

> > > ### Author Response · Authors · 2023-11-21
> > > **Thanks for the answer**
> > >
> > > Dear reviewer t4AA,
> > >
> > > Thank you for answering. Your comments were very helpful in improving our manuscript. We additionally provide the results of other methods like Bayesian optimization (GPBO) and a genetic algorithm (Graph GA) to compare with RL-based methods in molecular optimization.
> > >
> > > |  | GPBO | Graph GA | REINVENT | REINVENT + SRT |
> > > | --- | --- | --- | --- | --- |
> > > | Avg. AUC | 0.609 | 0.619 | 0.613 | **0.633** |

---

### Official Review · Reviewer_zPsa · 2023-11-04

**Soundness:** 3 good
**Presentation:** 3 good
**Contribution:** 2 fair
**Rating:** 6
**Confidence:** 4

**Summary:**

The paper is motivated by the observation that in various black-box combinatorial optimization problems, exploring the solution space with reinforcement learning can be very costly due to the limited availability of high cost of function evaluations. To address this, the authors notice that there can be multiple symmetric solutions with exactly the same reward (or cost) (e.g., by shifting the original solution, permuting it, etc.). They then propose symmetric replay training, a technique that consists in replaying symmetrically transformed high-reward samples, so that it can better explore the under-explored regions in the symmetric space. For this purpose, we do not need additional interactions with the environment, as the reward (or cost) function in the symmetric configuration is equal to the original one. The authors experimentally assess their approach on the traveling salesman problem, molecular optimization and hardware design. They find that symmetric replay training consistently improves the sample efficiency, often by large margins.

**Strengths:**

1. The proposed technique of symmetric replay training is simple yet effective. Furthermore, the paper is generally clear and easy to follow.
2. The experimental evaluation covers three distinct settings: a synthetic one (TSP) as well as two real-world scenarios)hardware design and molecular optimization). In all cases, symmetric replay training significantly improves sample efficiency. Furthermore, the authors try their technique with multiple RL algorithms, since the framework is independent of the used RL algorithm.
3. The positive results even compared to state-of-the-art methods (e.g., in practical molecular optimization) demonstrate the validity of the proposed approach.
4. The task, experimental settings, trailing process, loss functions and hyperparameters for each of the three settings is discussed in detail, both in the main text as well as in the appendix. I like the fact that the synthetic problems are compared to other sample-efficient techniques such as Syn-NCO.

**Weaknesses:**

1. Overall, the novelty of this work is not very significant. The symmetric replay training idea is rather straightforward and the same holds for the 2-phase algorithm. On the theory front, there are few theoretical results and insights, but at least the experimental evaluation is quite extensive and the results are positive.

2. There are other architectures with inductive bias for symmetries, such as permutation-invariant neural networks by Tang et al, besides the DevFormer. Since this paper is mostly experimental, it would have been great to assess how the proposed technique performs on top of such architectures that already incorporate the symmetric inductive bias. The experiments with the DevFormer in Table 2 and Figure 6 suggest that we can get additional benefit by applying symmetric replay training on top of such symmetric architectures. It would be interesting to understand if this is generally true with other architectures, too. In particular, even if the policy (and possibly value) networks already incorporate the symmetry in their architecture, is it possible for symmetric replay training to give additional benefit, and why would that happen?

3. The ablation study in Figure 4a is a bit unclear to me. How is it in principle possible to do simultaneous updates, given that we must first collect the high-reward samples in Phase 1? I think the idea of a 2-phase algorithm and alternating steps is inevitable, given that we must fist isolate the high-reward samples before applying the symmetric replay training. The authors could explain this part better to avoid confusion. Did they for example only store the high-reward samples in the replay pool for the simultaneous update, and then performed simultaneous updates once the replay buffer was sufficiently large?

**Questions:**

1. Have the authors tried other symmetric architectures besides the DevFormer? Does symmetric replay training on top of symmetric policy and value network yield additional benefit, and why? The proposed technique could complement other ideas such as symmetric NNs, so their combined impact would be interesting to understand.

2. Can the authors explain how simultaneous updates were performed in Figure 4a? Did they first conduct standard RL to fill the replay buffer with high-reward samples, and then switched to simultaneous updates? This was unclear from the text.

3. The paper emphasizes the fact that evaluating the black box function is an expensive operation. However, Table 8 shows that the proposed method can outperform other sample-efficient algorithms such as Sym-NCO in the synthetic settings, where function evaluation is fast and cheap. In that case, it seems that symmetric replay training could be positioned as a general technique for improved sample efficiency that can be used not only with black-box optimization but also more generally. The authors state "broad spectrum of CO problems and methods" and "a new generic DRL method" in the main text, so perhaps symmetric replay training is positioned as a general-purpose technique. On the other hand, the black-box nature (with expensive function evaluations) is mentioned several times and even in the title. It would make sense for the authors to disambiguate this point and position their method very clearly (i.e., as a general-purpose algorithm or a technique that is better suited for black box optimization with expensive function evaluations).

---

> ### Author Response · Authors · 2023-11-17
> **Response for the reviewer zPsa**
>
> Thanks for the valuable comments.
>
> **W1: the novelty of this work is not very significant.**
>
> As you commented, our algorithm is both straightforward and effective. Our key novelty beyond our simplicity lies in proposing new replay training methods, which can also be applied to both off-policy and on-policy reinforcement learning in combinatorial optimization. Up until now, prior replay training methods have primarily been designed for off-policy reinforcement learning. Please take into consideration the significance of this contribution.
>
>
> **W2, Q1: why does applying SRT on top of such symmetric architectures have additional benefits?**
>
> Solution symmetries, defined as situations where multiple trajectories can lead to identical solutions, are challenging to incorporate directly into architectural structures. Therefore, loss optimization is needed to induce solution symmetries. In DevFormer, they introduce an andditional loss term to consider solution symmetries; however, these symmetries are not induced in a structured way. On the other hand, GFlowNet introduces a new kind of loss function that considers solution symmetries by modeling the sequential decision-making in CO on a directed acyclic graph (DAG) and considering flows on DAG as a policy. Our two-step learning procedure can be used along with these DRL methods, and SRT can further induce solution symmetries effectively.
> It is noteworthy that our algorithm can also easily cooperate with symmetric architectures like SE3 Transformer [1] and EGNN [2], which exhibit architectural symmetries in state-level representation (e.g., rotation of an image yields the same representation). Since these models (including the permutation-invariant NN) do not consider solution symmetries related to the action trajectories, further learning processes like SRT are required to induce solution symmetries.
>
> **Answer for W3 and Q2: how simultaneous updates were performed.**
>
> In our two-step training, the model is updated to minimize the policy gradient loss in Step A. Then, the model is updated to minimize imitation loss in Step B. These updates are conducted alternatively. On the contrary, in the ablation, after calculating RL loss, we collect the high-rewarded samples using greedy rollout without model updating. We compute $L_{SRT}$ using the high-rewarded samples and then update the model with $L = L_{RL}+L_{SRT}$ at once. To clarify this, we revised the manuscript.
>
> **Answer for Q3: overemphasizing black-box nature (SRT is positioned as a general-purpose technique)**
>
> We agree that our SRT can give general-purpose replay training on CO. Our intention was that sample efficiency matters to real-world applications of CO, such as hardware optimization and molecule optimization. We revised the manuscript based on your suggestion.
>
> ----
> **Reference**
>
> [1] Fuchs, Fabian, et al. "SE(3)-Transformers: 3D roto-translation equivariant attention networks." NeurIPS, 2020.
>
> [2] Satorras, Vıctor Garcia, Emiel Hoogeboom, and Max Welling. "E (n) equivariant graph neural networks." International conference on machine learning. PMLR, 2021.

---

> ### Author Response · Authors · 2023-11-21
> **Gentle reminder for further discussion**
>
> Dear Reviewer zPsa,
>
> Thank you for sharing your thoughtful comments. We kindly encourage you to review our response to ensure that we have effectively addressed the concerns and questions raised in your comments. We are open to further discussion and would appreciate your feedback. If you have any additional questions or unresolved issues, please feel free to let us know.

---

> > ### Author Response · Authors · 2023-11-23
> > **Friendly Reminder: The discussion period will end soon!**
> >
> > Dear reviewer zPsa,
> >
> >
> > We wanted to send you a quick reminder that the end of the discussion is less than 6 hours away.
> > Please make sure that your concerns are properly addressed. Feel free to reach out if you have any last-minute questions or concerns.
> >
> > Regards, authors.

---

### Author Response · Authors · 2023-11-17
**General responses**

Thanks for all the valuable comments. We address each concern and question separately.

Overall, we have made revisions to our title and abstract to make the manuscript better align with our research aim. Our new title is **"Symmetric replay training: enhancing sample efficiency in deep reinforcement Learning for combinatorial optimization."** Please kindly check our revised manuscript, as well.

---

### Meta-Review · Area_Chair_KvAb · 2023-12-03

**Metareview:**

The paper proposes data augmentation (specifically in the form of symmetries on the action output) to improve RL-based combinatorial optimization. Experiments were conducted via a rough cross-product between methods (A2C, PG, PPO, Generative Training), architectures (Attention Model, Device Transformer, GFlowNet, RNN), and problems (TSP, Device Placement, Molecular Optimization). The data augmentation was shown to nearly unanimously improve scores, some marginally and some significantly.

There was a lengthy discussion between reviewers, with main points:
* In the initial phase of reviewing, a common issue was the scope of the work (i.e. whether to compare against non-RL based methods, such as Bayesian Optimization). The authors addressed this by modifying the emphasis to be focusing on RL-CO.
* The paper's simplicity and whether this is positive (general insight) or negative (lack of novelty, "just hyperparameter tuning"). The authors addressed this by claiming their work shows improvement across both on-policy and off-policy methods, which is a new insight.

If we look at precedence, note that studying symmetry-based data augmentation RL-CO is not new, and has been studied by previous papers [1,2,3], with some accepted and some rejected. The question is whether this new paper contributes to additional new insights unseen previously, especially against [3] which has already shown data augmentation improves performance for on-policy method.

The answer is yes: the paper's comprehensiveness across both off-policy and on-policy methods are convincing enough to suggest that symmetric augmentation practices in RL-CO should be universally used to improve performance. But these new contributions are not highlighted and ablated well in the paper (outside of simple raw numbers).

Thus I recommend borderline rejection for now. I strongly suggest that the authors take more effort to emphasize these novel contributions over previous works and resubmit to another venue.

[1] https://openreview.net/forum?id=a_yFkJ4-uEK (Rejected by ICLR 2023)

[2] https://arxiv.org/abs/2202.08396 (ICML 2022)

[3] https://arxiv.org/abs/2010.16011 (NeurIPS 2020)

**Justification For Why Not Higher Score:**

This paper is incredibly borderline already, and there are no reviewers which strongly champion its acceptance. As mentioned, there are previous works already which touch upon the augmentation topic, and the new contributions for data augmentation in RL-CO are not sufficiently expanded upon.

**Justification For Why Not Lower Score:**

N/A

---

### Decision · Program_Chairs · 2024-01-16

Reject